# Surface chemistry governs cellular tropism of nanoparticles in the brain

Eric Song[1],*, Alice Gaudin[1],*, Amanda R. King[1], Young-Eun Seo[1], Hee-Won Suh[1], Yang Deng[1], Jiajia Cui[1], Gregory T. Tietjen[1], Anita Huttner[2] & W. Mark Saltzman[1]

Nanoparticles are of long-standing interest for the treatment of neurological diseases such as glioblastoma. Most past work focused on methods to introduce nanoparticles into the brain, suggesting that reaching the brain interstitium will be sufficient to ensure therapeutic efficacy. However, optimized nanoparticle design for drug delivery to the central nervous system is limited by our understanding of their cellular deposition in the brain. Here, we investigated the cellular fate of poly(lactic acid) nanoparticles presenting different surface chemistries, after administration by convection-enhanced delivery. We demonstrate that nanoparticles with 'stealth' properties mostly avoid internalization by all cell types, but internalization can be enhanced by functionalization with bio-adhesive end-groups. We also show that association rates measured in cultured cells predict the extent of internalization of nanoparticles in cell populations. Finally, evaluating therapeutic efficacy in an orthotopic model of glioblastoma highlights the need to balance significant uptake without inducing adverse toxicity.

[1] Department of Biomedical Engineering, Malone Engineering Center, Yale University, New Haven, Connecticut 06510, USA. [2] Department of Pathology, Yale University, New Haven, Connecticut 06520, USA. * These authors contributed equally to this work. Correspondence and requests for materials should be addressed to W.M.S. (email: mark.saltzman@yale.edu).

Recent developments in personalized medicine and drug delivery have produced exciting opportunities in oncology[1]. Despite these advances, as well as major progress in the development of new chemotherapeutics and improved surgical techniques, prognosis for individuals with high-grade glioma, such as glioblastoma multiforme (GBM), remains poor, with a median survival of 15 months[2]. Among the barriers preventing effective treatment of brain tumours, the blood–brain barrier (BBB) is the most prominent[3], hampering the achievement of relevant therapeutic concentrations of drug in the tumour mass without inducing systemic cytotoxicity. More generally, the BBB inhibits effective treatment in brain tumours, neurodegenerative diseases, stroke and every major condition that afflicts the central nervous system (CNS).

Nanomaterials have long been proposed as carriers to facilitate the entry and delivery of agents into the brain[4–6]. Long circulating nanoparticles (NPs), such as those decorated with polyethylene glycol (PEG) on their surface, have been proposed as a way to enhance brain penetration through the BBB. However, despite their so-called 'stealth' properties (referring to their resistance to opsonization and to elimination by the reticuloendothelial system[7]), usually <1% of the injected dose gains access to the brain tissue after systemic administration[5,8]. Transport through the nasal epithelium can be more efficient, although the small surface area and anatomical location tend to limit the value of this method[9]. Local infusion directly into the brain, or convection-enhanced delivery (CED)[10], can be used to slowly introduce large volumes of NPs into the cerebral interstitium, allowing the NPs to reside in the brain parenchyma while slowly releasing encapsulated agents over prolonged periods[11,12]. It has been suggested that to be efficient when administered by CED, NPs need to penetrate readily through the brain pores, or be 'brain-penetrating'[12–14]. In one approach, dense PEG surface coating of NPs has been used to increase the distribution of particles in the brain after CED[15], extending stealth properties to the brain interstitial space by preventing adhesive trapping in the extracellular matrix. However, even with brain-penetrating NPs, varying degrees of survival benefits in animal models were achieved[11,12,16], suggesting that other variables likely influence biological activity.

Previous studies of NP delivery in the brain have evaluated the macro-level interaction of NPs with tissues, by studying characteristics such as accumulation in the brain after systemic administration[17,18], and volume of distribution after CED[12,15,16], and relating these features to survival benefit. More recently, the influence of a tumour mass on the macroscopic pattern of NP distribution was investigated[19]. Yet, there are few studies on how surface properties of NPs influence association with particular cell types once they enter the brain interstitium.

In this report, we studied NP interaction with cells, or cellular tropism, in the context of the brain/tumour cellular microenvironment. Poly(lactic acid) (PLA) based NPs presenting different surface chemistries were administered into the rat brain by CED. We observed that NPs decorated with 'stealth' groups were able to avoid internalization by all cell types, including tumour cells. Internalization could be restored and enhanced by functionalization with bio-adhesive end groups. We also showed that in vitro association rates measured in cultured cells could predict the extent of NPs internalization by the different cell populations in vivo. Finally, the evaluation of the therapeutic efficacy of the different formulations loaded with the potent cytotoxic drug Epothilone B (EB) in an orthotopic model of glioblastoma underlined the need to balance significant uptake without inducing adverse toxicity to healthy cells. We believe that a better understanding of cellular fate for NPs after brain delivery will speed the development of more effective NP-based delivery systems for the treatment of brain diseases.

## Results

**Brain penetrating PLA-based formulations**. We focused our attention on a widely studied polymer system—PLA—and investigated how surface modifications of PLA NPs influence cellular tropism in the brain. Composed of one of the few degradable polymers approved by the Food and Drug Administration for medical applications[20], PLA NPs have been extensively studied because of their biodegradability and versatility, including for the treatment of neurological diseases. Four PLA-based NP formulations were compared (Fig. 1a). PLA–PEG NPs are in advanced clinical trials for systemic delivery of chemotherapeutics[21], however, PEG itself presents the disadvantage of a non-biodegradable main chain, and the possible induction of an anti-PEG immune response[22]. As an alternate, hyperbranched glycerol (HPG) can also be used as a surface coating of PLA NPs: PLA–HPG NPs produce a greater stealth effect than PLA–PEG, avoid recognition by phagocytic cells, and provide multiple functionalizing sites.[23] PLA–HPG NPs have the additional advantage that they can be turned into bio-adhesive NPs[24], by converting the vicinal diols of the HPG into aldehydes (-CHO), obtaining PLA–HPG–CHO NPs. Each of these four NP formulations (PLA, PLA–PEG, PLA–HPG and PLA–HPG–CHO NPs) were successfully engineered to fit the criteria to be brain penetrating[12], as they presented a diameter <100 nm when observed by TEM (Fig. 1b) and a hydrodynamic diameter <150 nm when measured by dynamic light scattering (Fig. 1c), had a neutral or negative surface charge (Fig. 1d), and were non-aggregating after incubation in artificial cerebrospinal fluid at 37 °C for up to 24 h (Fig. 1e). Particles were fluorescently labelled using the DiA dye, and engineered to provide comparable fluorescence intensities (Supplementary Fig. 1). We have previously evaluated the retention of the Di dyes by performing in vitro release studies, showing that the dye is not escaping PLA-based particles, even in the presence of very high concentrations of proteins[25].

As expected for brain-penetrating NPs, the four formulations distributed widely and homogeneously when introduced into rat brains by CED, reaching distribution volumes of $\sim 40\,mm^3$ (Fig. 2a,b; Supplementary Movie 1). The different NPs were also administered to rats bearing GFP expressing RG2 tumours grown for 7 days. All formulations yielded volumes of distributions (Fig. 2c) in the same order of magnitude as those observed in brains without tumours, in accordance with previous results showing that the presence of a tumour, whatever its size, does not significantly modify the particle volume of distribution[26]. However, the distribution of the particles appeared less homogeneous than in the healthy brain (Fig. 2d), likely due to the presence of the tumour mass.

**Cellular tropism in the healthy brain**. The brain environment is complex, with different cell types that are intimately and reciprocally linked to each other, establishing an anatomical and functional neurovascular unit[27], which ensures correct brain functions[28]. We chose to focus our attention on three cell types known to have crucial functional roles: neurons that support information transmission; astrocytes that perform many active functions including structural and metabolic support of neurons; and microglia that represent the main cellular immune defense in the brain. Four hours after introduction of particles into the brain interstitium, the injected hemisphere was processed (Supplementary Fig. 2a) and analysed by flow cytometry (Supplementary Figs 2b and 3) to measure cellular tropism of

the different particle types. The three cell populations were identified with specific intracellular markers (astrocytes, microglia or neurons identified by the markers glial fibrillary acidic protein (GFAP), Iba-1 and NeuN, respectively). For each formulation and each cell type, three pieces of information were extracted from the fluorescence-activated cell sorting data (see Methods section for detailed procedure): first, the percentage of cells positively internalizing NPs, which was determined by measuring the population of cells shifting away from the control population (Fig. 3a, red area, Supplementary Fig. 4), second, the number of NPs internalized by this population, which was measured by the mean fluorescence intensity (MFI; Fig. 3a,b), and third, the relative amount of NPs in each cell population, which takes into account the relative abundance of cells within the brain, and is expressed as a percentage of total number of NPs associated with cells (Fig. 3c). The total NP association is depicted as the area of the pie charts in Fig. 3c. Compared to the reference formulation, PLA NPs, the total NP uptake was substantially lower for PLA–PEG and PLA–HPG NPs, but significantly higher for PLA–HPG–CHO NPs (Fig. 3b,c). More specifically, compared to PLA NPs,

for which the percentage of cells that shifted out of their control population for all cell types was ∼4% (Supplementary Fig. 4), administration of PLA–PEG and PLA–HPG NPs resulted in lower percentages of cells that shifted (∼2.7 and 1.6% respectively), and administration of PLA–HPG–CHO NPs produced greater population shifts of ∼5.2% (Supplementary Fig. 4). Interestingly, PLA–PEG and PLA–HPG NPs distributed relatively evenly between all three cell types (Fig. 3b,c), whereas PLA–HPG–CHO NPs presented a preferential uptake by microglia cells and decreased uptake by neurons, similar to PLA NPs. To examine time-dependent cellular tropism, we quantified cellular association 24 h after particle administration (Fig. 3c, Supplementary Fig. 5). Each NP formulation was internalized more abundantly after 24 h compared to 4 h with the extent of increase being significantly influenced by the NP surface properties. PLA–PEG and PLA–HPG NPs presented a 1.4 and 2.3-fold increase, while internalization of PLA–HPG–CHO NPs was increased even more than the PLA NPs (four- and threefold increase respectively; Fig. 3c). These latter two NP formulations displayed different patterns regarding total population shift after

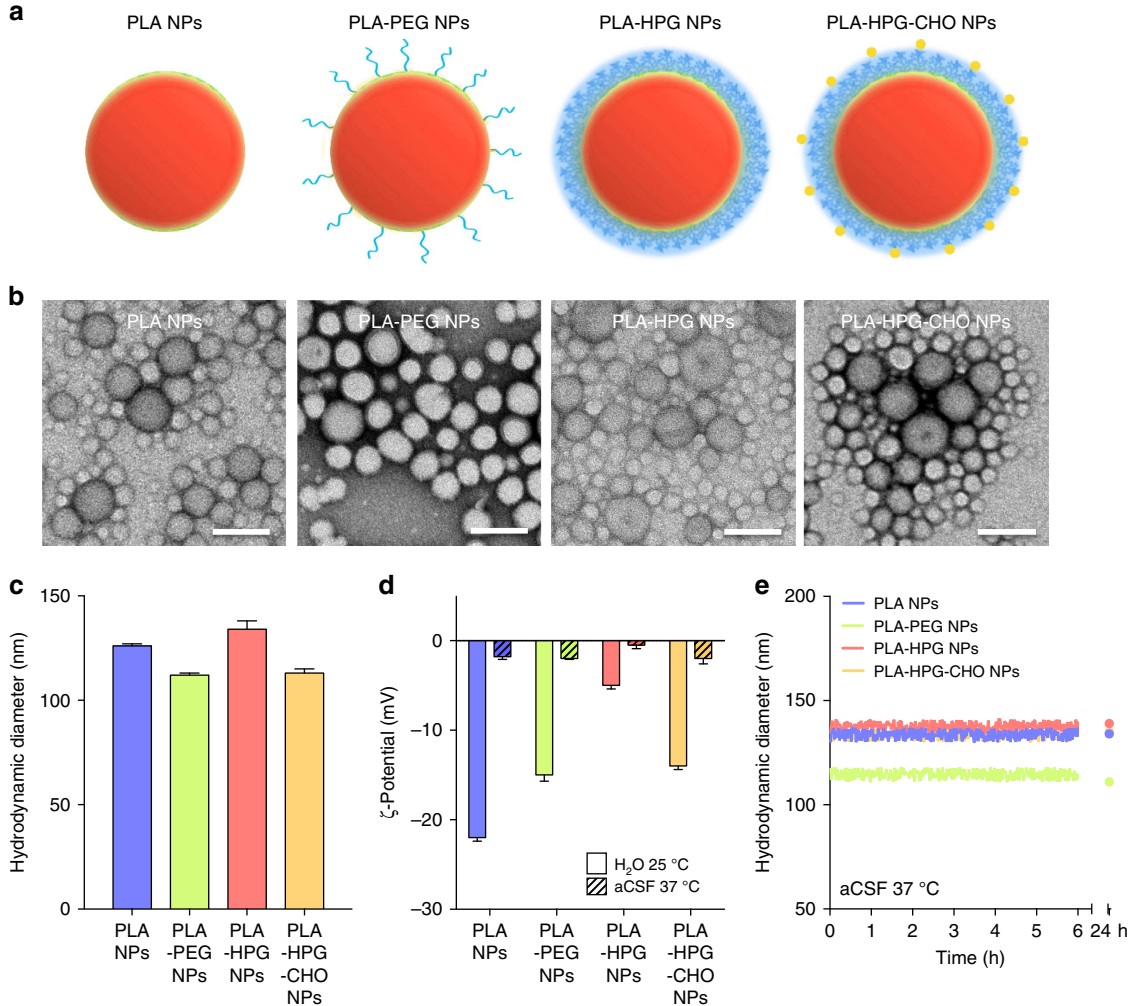

**Figure 1 | NP physico-chemical characterization using electron microscopy and dynamic light scattering.** (**a**) Schematic of PLA-based NPs with different surface coatings. (**b**) Particle morphology and population was imaged using TEM after staining with uranyl acetate, with each image corresponding to the image above in Fig. 1a (representative image of $N = 3$ biological replicates, scale bar = 100 nm). Particle characterization with dynamic light scattering (**c**) and laser doppler anemometry (**d**) displayed similar hydrodynamic diameters and zeta potential respectively for all particle types. Size analysis was conducted in water (**c**) and zeta potential was measured in water and in artificial cerebrospinal fluid (aCSF), showing the neutralization of all particle surface charge in aCSF (results are presented as mean ± s.d. of $N = 3$ biological replicates) (**d**). (**e**) Particle size was shown to be stable in 37 °C aCSF without measurable aggregation for up to 24 h (representative graph of $N = 3$ biological replicates).

24 h (Supplementary Fig. 4). For PLA NPs, a large percentage of the cell population (22–28%) internalized a significant but relatively low amount of particles, whereas the PLA–HPG–CHO NP MFIs were increased in a smaller population of cells (6.3–8.1%), taking up a large number of particles. These observations suggest that although global uptake was higher for PLA–HPG–CHO NPs compared to PLA NPs, it affected a smaller population of cells. Notably, as more NPs become associated with cells from 4 to 24 h, the distribution among the different cell types remained the same for all NPs formulations (Fig. 3c).

We further characterized the NP distribution among cells using confocal microscopy (Fig. 4 and Supplementary Figs 6 and 7). Staining for neurons (NeuN), no significant differences were observed in terms of cellular morphology (Supplementary Figs 6 and 7). In astrocytes (GFAP), PLA–HPG–CHO NPs produced an up-regulation of GFAP protein, characteristic of reactive astrocytes[29] (Fig. 4a–h). Finally, in the case of microglia (Iba-1), significant morphological differences were observed (Fig. 4i–p). These changes were dependent on the NP type: similar to what was observed for PLA NPs, the introduction of PLA–HPG–CHO NPs led to an amoeboid shape, typical of an activated state[30] (Fig. 4i,l), whereas after introduction of PLA–PEG or PLA–HPG NPs, microglia retained a ramified shape typical of an inactivated state (Fig. 4j,k), similar to untreated brain (Supplementary Fig. 6).

Confocal imaging also confirmed an increased uptake of all particle types 24 h after introduction (Fig. 4e-h and m-p), with the PLA–PEG and PLA–HPG NPs being internalized the least (Supplementary Fig. 7b). For all NP types, the background in the particle channel was decreased and the amount of NPs in the perinuclear space of cells was increased at 24 h compared to 4 h (Supplementary Fig. 8), with the highest intensity observed in PLA–HPG–CHO NP treated brains (up to 75% of the PLA–HPG–CHO NPs were found in the perinuclear space after 24 h, Supplementary Fig. 8d). The significant increase of NPs in the perinuclear space between 4 and 24 h (Supplementary Fig. 8) strongly suggests an active internalization of all NPs type, rather than a passive association with the cellular membrane. To explore this in more detail, we performed an *in vitro* experiment using representative cell lines of the different cell types found in the brain (TNC1 cells for astrocytes, BV-2 cells for microglia and N27 cells for neurons), incubating them with the different formulations at 37 or 4 °C to assess for active uptake. For all cell types and all formulations, the uptake at 4 °C did not exceed 10% of the uptake at 37 °C (Supplementary Fig. 9), confirming an active internalization of all NPs formulation by the brain cells.

Confocal images also confirmed the observations regarding PLA and PLA–HPG–CHO NP internalization patterns (Supplementary Fig. 4): while PLA NPs appeared to be

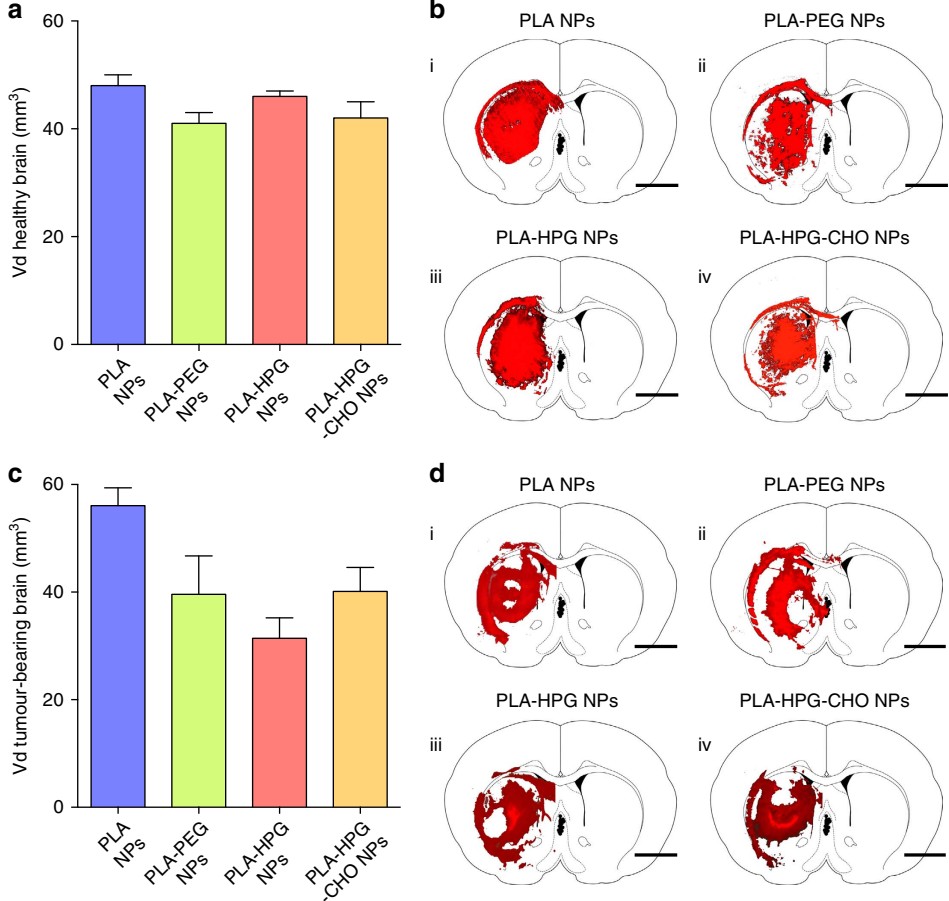

**Figure 2 | Volume of distribution after administration by CED.** (**a**) When fluorescently labelled particles were infused in healthy Fischer 344 rats via CED, they displayed similar volumes of distribution (Vd; results are presented as mean ± s.d. of N = 3 biological replicates). (**b**) i–iv, for all particle types, distribution in the healthy brain was homogeneous through the whole caudate (representative image of N = 3 biological replicates, scale bar, 4 mm). (**c**) When infused in Fischer 344 rats bearing RG2 tumours grown for 7 days, the different formulations displayed similar volumes of distribution (Vd), except for the PLA NPs that appeared to distribute more widely (results are presented as mean ± s.d. of N = 3 biological replicates). (**d**) i–iv, for all particle types, distribution pattern was more heterogeneous in the tumour-bearing brain compared to the healthy brain because of the presence of the tumour mass (representative image of N = 3 biological replicates, scale bar, 4 mm).

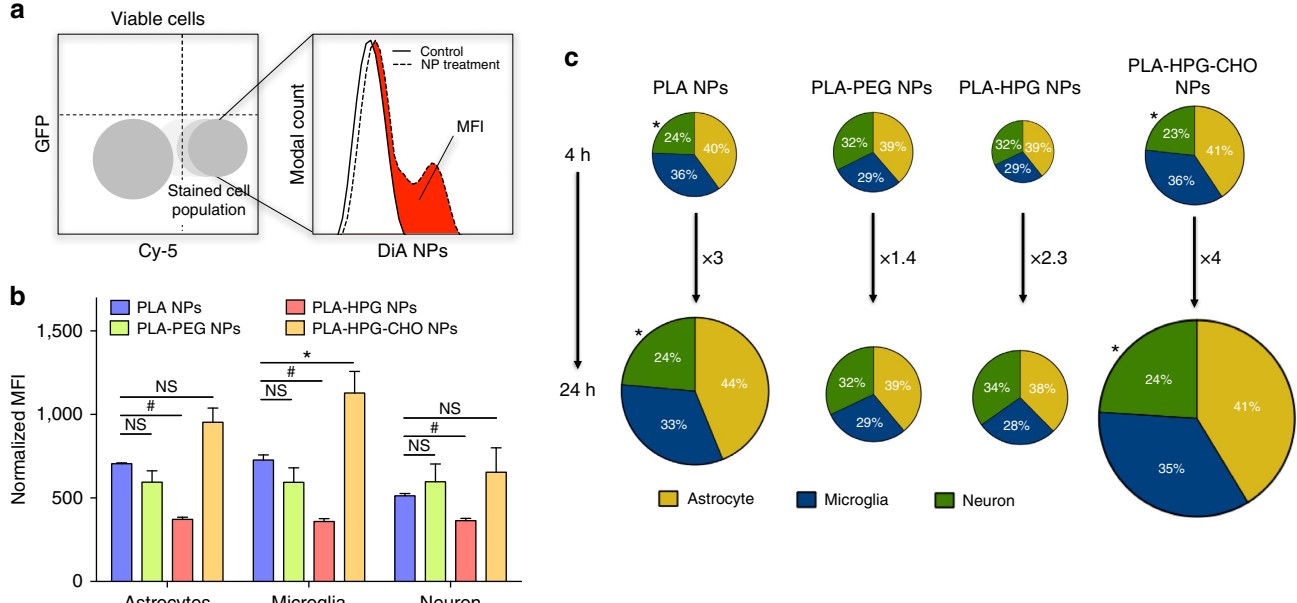

**Figure 3 | Cellular tropism of NPs 4 and 24 h after introduction into the interstitium of the healthy brain.** (**a**) Schematic of cell population shift and calculated MFI value, more detailed methods are outlined in the Method section and Supplementary Fig. 2. (**b**) Mean fluorescence intensity of each cell population in the DiA channel measured by flow cytometry (results are presented as mean ± s.d. of $N = 5$ biological replicates, experiments of same particle type were done on different days to ensure reproducibility of processing, two control brains were harvested each day, statistical analysis was performed using a two-sided Student's $t$-test, $*P < 0.05$, $\#P < 0.005$). The particles were delivered at a concentration of 50 mg ml$^{-1}$ and the MFI was normalized by the relative loading of the dye (Supplementary Fig. 1). (**c**) Absolute amount of fluorescence at 4 and 24 h for all particle types was derived by multiplying the MFI by the relative number of cells in each population, also measured by flow cytometry (refer to methods, statistical analysis was performed using a two-sided Student's $t$-test, $*P < 0.05$). Total area of the pie charts denotes the sum of the absolute fluorescence within the three cell populations, representing the total NP uptake by these cells, and each slice gives the relative particle uptake for each cell population (Supplementary Table 1).

internalized homogeneously by a large number of cells, fewer cells took up the PLA–HPG–CHO NPs but to a greater extent (Supplementary Fig. 7b). Of note, after 24 h, we observed an up-regulation of GFAP proteins in brains administered with PLA NPs (Fig. 4e), and activated microglia in PLA–PEG NP treated brains (Fig. 4n). On the other hand, PLA–HPG NPs did not induce activation of microglia (Fig. 4o) nor did they increase the presence of reactive astrocytes (Fig. 4g), even 24 h after introduction into the brain interstitium. Overall, these results show that in the healthy brain, PLA–PEG and PLA–HPG NPs are internalized substantially less compared to PLA NPs, and conversion of diols on HPG to aldehyde groups reverse and increase uptake in all cell types.

**Cellular tropism in the tumour-bearing brain.** Tumour cells have been shown to strongly influence cellular interactions within the brain microenvironment, forming niches that allow for tumour protection and proliferation[31,32]. They are also able to manipulate their cellular environment via secretion of proteins and display of cell surface ligands that promote tumour growth[33]. We first analysed the cellular composition of the brain to quantify the cell populations in the tumour-free brain, compared to brains 7 or 8 days after introduction of RG2-GFP cells (Fig. 5). At 7 or 8 days of growth, tumour cells accounted for 13 and 18%, respectively, of the total cell population in the hemisphere (Fig. 5a), confirming the fast development of RG2 tumours[34]. The fraction of neurons was constant among the different brains (9–10%), and consistent with literature values[35]. The fraction of microglia cells was slightly increased in the presence of the tumour, likely due to the recruitment of tumour-associated macrophages (TAMs)[36]. Finally, a sub-population of tumour cells

(accounting for ∼38% of the tumour population) was positive for GFAP, reflecting the astrocytic origin of RG2 tumours[37]. Imaging of the tumour-bearing brain demonstrated the presence of activated microglia/TAMs within the tumour bulk (Fig. 5b), a strong astrogliosis at the periphery of the tumour (Fig. 5f), and the total absence of neurons inside the tumour bulk (Fig. 5h). Interestingly, within the tumour bulk, all GFAP-positive cells were also GFP positive (Fig. 5e), suggesting that the majority of non-tumoral cells present inside the tumour were microglia and TAMs. These observations were consistent with previous reports on rat and human GBM[30,38].

Infusion of NPs was performed after 7 days of tumour growth. Four hours after infusion, a significant fraction of NPs was associated with tumour cells (Fig. 6a), and this fraction varied with particle chemistry. While 32% of the tumour cells were associated with PLA NPs, only 11% were associated with PLA–PEG or PLA–HPG NPs, and up to 76% were associated with PLA–HPG–CHO NPs. Despite the varying amounts of uptake, all three particle types presenting surface modification (PLA–PEG, PLA–HPG and PLA–HPG–CHO NPs) displayed preferential uptake by tumour cells compared to other cell types (Fig. 6b-e), while PLA NPs were equally internalized by tumour cells and microglia cells (Fig. 6b). Interestingly, the HPG surface modification was more efficient at decreasing microglia and neuron uptake compared to PEG, allowing for the highest specificity towards tumour cells (Fig. 6c,d, Supplementary Fig. 10), although the total uptake for both stealth formulations (PLA–PEG and PLA–HPG NPs) was low. Confocal imaging 4 h after introduction of NPs into the brain confirmed that NPs were internalized by tumour cells, microglia/TAM and GFP/GFAP-positive tumour cells inside the tumour bulk (Fig. 6j,k), with significant uptake by activated microglia and reactive astrocytes at

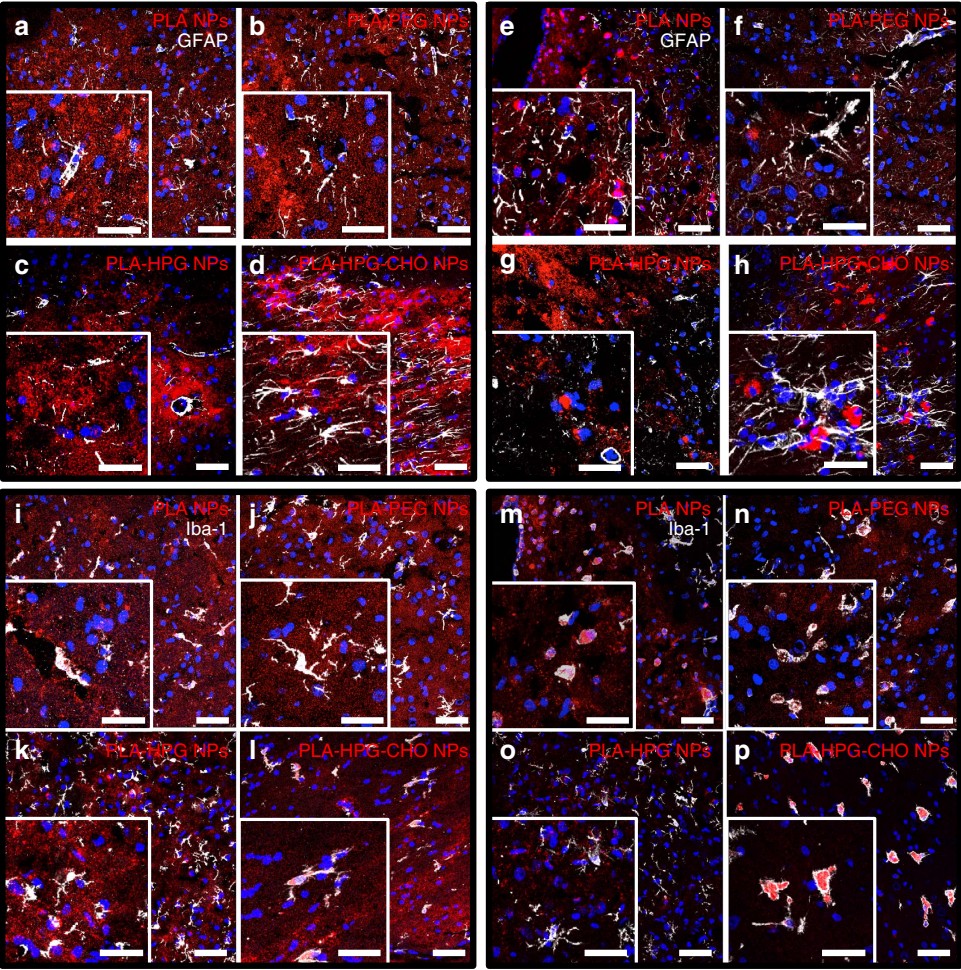

**Figure 4 | Confocal images illustrating cellular tropism of NPs 4 and 24 h in the healthy brain.** (**a**–**d**) Brain slices stained for astrocytes (GFAP in white), showing that PLA–HPG–CHO NPs (red) produced an up-regulation of GFAP protein, characteristic of reactive astrocytes (**d**), while PLA, PLA–PEG (red) and PLA–HPG NPs (red) did not (**a,b,c**) at 4 h. (**e**–**h**; GFAP in white) After 24 h, all particles except PLA–HPG NPs (red) (**g**) induced an up-regulation of GFAP proteins. (**i**-**l**) Brain slices stained for microglia (Iba-1 in white), showing that PLA (red) and PLA–HPG–CHO NPs (red) activated microglia (**i,l**, microglia present mostly in amoeboid shape), while PLA–PEG (red) and PLA–HPG NPs (red) did not (**j** and **k**, microglia retain their ramified state) at 4 h. (**m**-**p**; iba-1 in white) After 24 h, all particles except PLA–HPG NPs (red) (**o**) activated microglia, with PLA–HPG–CHO NPs (red) showing the largest amount of particles in the perinuclear space (**p**, Supplementary Fig. 8). For each particle type and staining, the image is representative of three slides from one animal. All images were stained with DAPI (blue) for nuclear visualization. (Scale bar, 20 μm for zoomed in images and 50 μm for larger images).

the tumour periphery (Supplementary Fig. 11). The extent of uptake of all NPs types was significantly increased 24 h after their introduction in the interstitial space (Fig. 6b–i, Supplementary Fig. 12), especially for tumour cells. Once again, the fraction associated with NPs depended on their surface properties: increasing to 66% for PLA NPs, the fraction was increased only to 18% and 16% for PLA–PEG and PLA–HPG NPs, respectively, and to 87% for PLA–HPG–CHO NPs. Overall, normalization of total uptake for all particle types and conditions (healthy brain versus tumour-bearing brain, 4 versus 24 h) showed that compared to PLA NPs, PLA–HPG–CHO NPs displayed the highest internalization in all conditions, while PLA–PEG and PLA–HPG NPs presented the lowest uptake level (Supplementary Table 1). The higher internalization for PLA–HPG–CHO NPs extended to all cell types, including healthy cell populations (astrocytes, microglia and neurons; Supplementary Fig. 13). These observations were further confirmed by confocal microscopy (Supplementary Fig. 14), and similarly to the healthy brain, the amount of particles in the extracellular space was reduced at 24 h compared to 4 h, whereas particles appeared concentrated in the perinuclear space (Supplementary Fig. 8).

***In vitro* prediction of *in vivo* cellular tropism.** For many brain diseases, the cellular target is known: tumour cells and microglia in the case of brain tumours[39], microglia and neurons in the case of Alzheimer's disease[40], or astrocytes in the case of amyothrophic lateral sclerosis[41]. NPs are promising candidates for delivery of therapeutic agents in these settings, but there is no reliable way to determine what NP compositions are most efficient for each set of cellular targets. An *in vitro* screening method to identify the most relevant NP properties would be helpful. Here, we correlated our observations of NPs association with cells in the brain with kinetic measurements of particle uptake in cultured cells (Fig. 7a, Supplementary Fig. 15). Since they are not decorated with any active-targeting ligands, we made the assumption that all particles were experiencing a similar uptake mechanism, likely utilizing non-specific endocytosis pathways[42]. Also, since the four particle formulations used in this study were engineered to have similar size, surface charge and stability in physiological conditions, along with comparable volumes of distribution when introduced into the brain (Figs 1 and 2), we predicted that their cellular uptake would be governed by the rate of association between the NPs and the cells, likely due

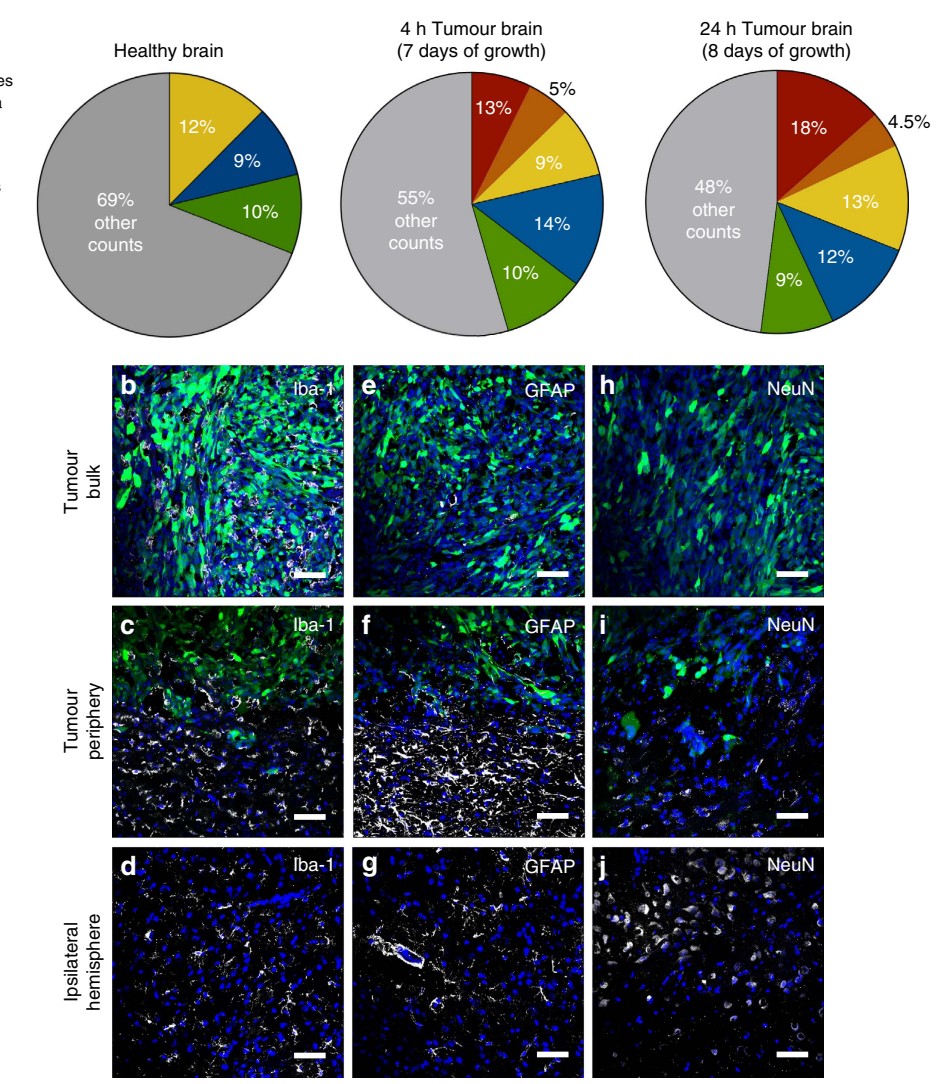

**Figure 5 | Cellular composition of rat brains with and without orthotopic tumours.** Cell populations were determined using flow cytometry in the healthy brain and the tumour-bearing brain, after 7 and 8 days of tumour growth following the implantation of 250,000 RG2-GFP cells ($N = 6$ biological replicates). (**a**) Pie charts of cellular content from fluorescence-activated cell sorting analysis of healthy and tumour-bearing brains. The other counts include cells, but also debris that could not be gated out successfully. The amount of neurons appeared constant between the different brains. The amount of microglia cells was slightly increased in the presence of the tumour, likely due to the recruitment of TAM. In the tumour-bearing brain, a small amount of tumour cells was positive for GFAP labelling, in accordance with the astrocytic origin of RG2 tumours (green). Representative confocal images of microglia (white) (**b–d**), astrocytes (white) (**e–g**) and neurons (white) (**h–j**) in the tumour bulk, around the periphery of the tumour and on the edge of the ipsilateral hemisphere. (**c,f**) High density of microglia and astrocytes border the tumour periphery and recruitment of activated microglia/tumour-associated macrophages is notable in the tumour bulk (**b**). For each cell type and area the image is representative of three slides from one animal. All images were stained with DAPI (blue) for nuclear visualization. (Scale bar, 50 μm).

to differences in the protein corona acquired by the NPs as they resided in the brain interstitium[42,43]. *In vivo*, the NPs were administered at a concentration where the system was not saturated in terms of particle concentration, so the MFI of each particle in a cell population at a given time point was used as a measurement of the rate of association of the NPs to a specific cell type (Fig. 3b, Supplementary Figs 5 and 12). *In vitro*, NPs were delivered to cells in culture at a concentration that saturated the system during the 24 h experiment, so the rate of association of the NP formulation to a cell type could be derived from a simple rate equation. Therefore, we hypothesized that there should be a correlation between the *in vivo* MFI values (Fig. 7c,d) and the rate of uptake measured *in vitro* (Fig. 7b) since both were measurements of the rate of association of particles to a cell type. Indeed, when a linear regression was applied to these two values,

we saw significant slopes in both healthy and tumour-bearing brains, for 4 and 24 h time points ($P < 0.001$). Fitting of the model to the data were more predictive at 4 h compared to 24 h, both in the healthy brain ($R^2 = 0.8232$ and 0.7504 at 4 and 24 h, respectively, Fig. 7e,f), and the tumour brain ($R^2 = 0.8397$ and 0.5387 at 4 and 24 h, respectively, Fig. 7g,h).

**In vivo therapeutic efficacy and toxicity.** We then evaluated if this more complete understanding of the interactions between the different NP types and the brain tissue was correlated with therapeutic outcome. Assuming that therapeutic efficacy against tumours is directly related to the absolute extent of internalization of NPs by tumours cells, the simplest interpretation of our results would be to expect the PLA–HPG–CHO NPs to be more

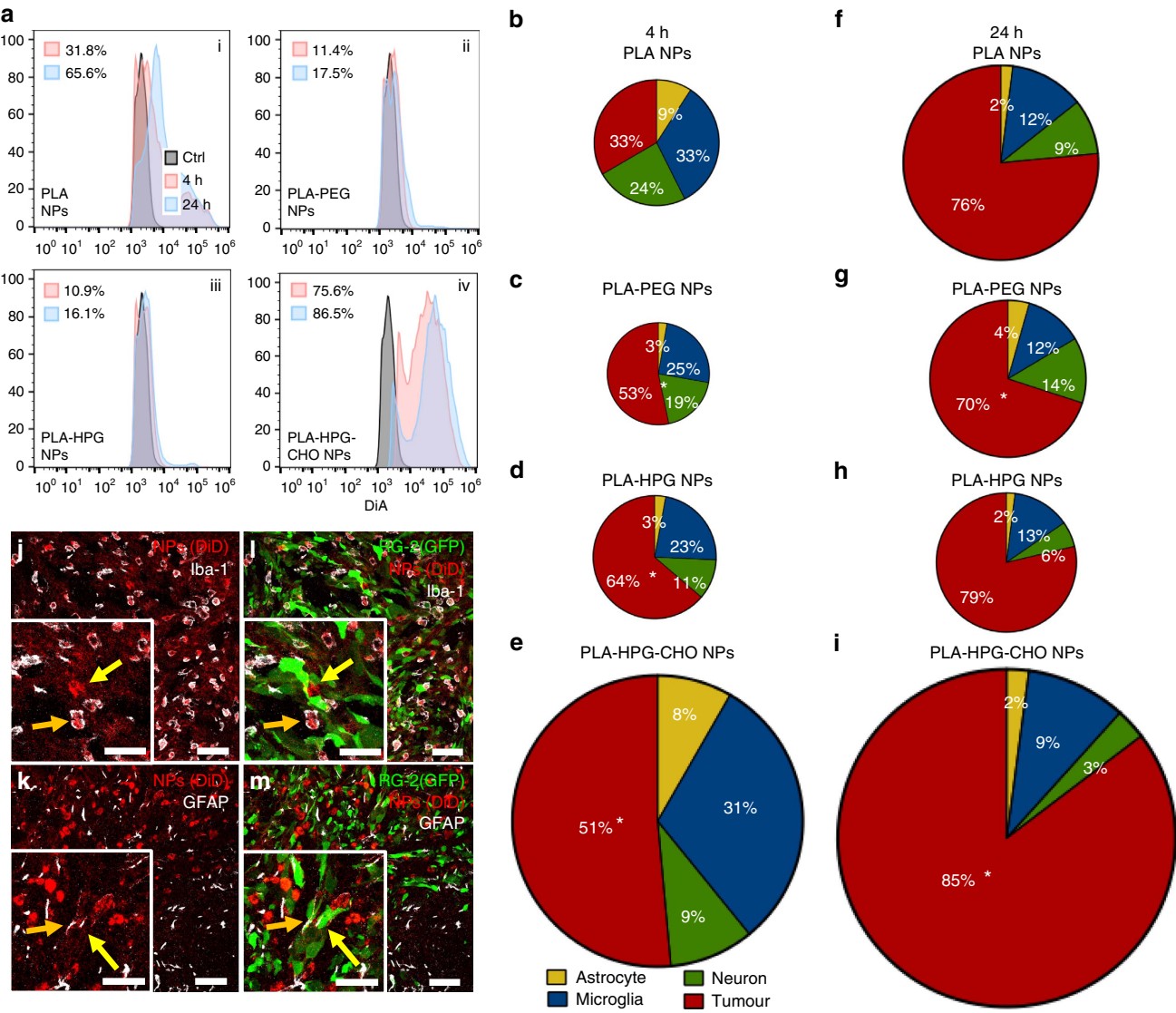

**Figure 6 | Cellular tropism of NPs 4 and 24 h after CED in the tumour-bearing brain.** (**a**) i–iv, representative histograms of population shift of tumour cells, after internalization of NPs loaded with DiA at 4 and 24 h (red and blue respectively). The per cent shift is an average of N = 5 biological replicates (experiments of same particle type were done on different days to ensure reproducibility of processing, two control brains were harvested each day) and the MFI of the shifted population is displayed in Supplementary Fig. 12. (**b–i**) Absolute amount of fluorescence was derived by multiplying the MFI by the relative number of cells in each population, also measured by flow cytometry. Total area of the pie charts denotes the sum of the absolute fluorescence within the four cell populations, representing the total NP uptake by these cells and each slice gives the relative particle uptake for each cell population. Change in uptake between 4 h (**b–e**) and 24 h time points (**f–i**) show markedly increased selective uptake by tumour cells compared to other cell populations. A statistical analysis was performed using a two-sided Student's t-test to compare the tumour fractions of each particle type to the reference formulation, PLA NPs (*P < 0.05). (**j,l**) Confocal image of microglia staining (white) 4 h after CED. Orange arrow indicates microglia/tumour-associated macrophage with perinuclear uptake of NPs. Yellow arrow indicates perinuclear uptake of NPs (red) by RG2-GFP cells (green). (**k,m**) Confocal image of astrocyte staining (white) 4 h after CED. Orange arrow indicates GFAP-positive RG2-GFP processes and yellow arrow indicates GFAP-positive RG2-GFP cell body with perinuclear NP uptake. For each particle type, the image is representative of three slides from one animal. (Scale bar, 20 μm for zoomed in images and 50 μm for larger images).

efficient than the PLA NPs, which would be more efficient than both stealth formulations (PLA–HPG and PLA–PEG NPs; Fig. 6, Supplementary Table 1). To test this hypothesis, the four formulations were engineered to encapsulate identical amount of EB, a potent chemotherapy drug, while retaining similar physicochemical characteristics (Supplementary Table 2). In particular, we observed that the four formulations presented the same pattern of EB release, with a burst release during the first hours of incubation, followed by a slow release over the following days of incubation (Fig. 8a). This suggests that the overall drug release pattern is mainly governed by the nature of the NP core, which is

PLA for all the NPs formulation, and is not dramatically influenced by surface modifications. However, after 24 h (Fig. 8a, insert), the PLA NPs had released 40% of their payload, the PLA–HPG and PLA–HPG–CHO NPs around 30%, and PLA–PEG NPs only 16%, suggesting that surface modifications can influence the initial release phase. Similarly, after 14 days, the PLA, PLA–HPG and PLA–PEG formulations released up to 80–90% of the drug, while the PLA–HPG–CHO NPs formulation released only 65%. We then evaluated those formulations in our orthotopic model of GBM. Seven days after RG2 tumours implantation, animals were treated by CED with one of the four

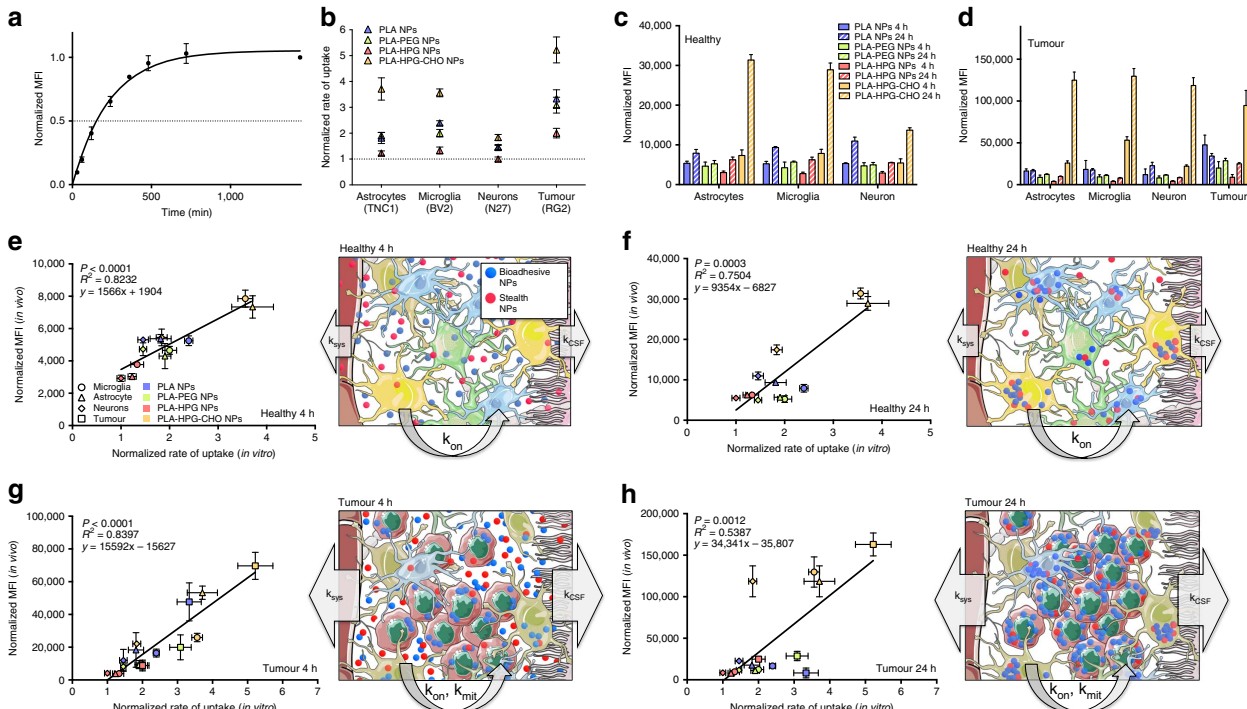

**Figure 7 | In vitro rates of uptake correlate with uptake of NPs in specific cell types in vivo.** (**a**) Fitting of kinetic of association equation to NP *in vitro* uptake data. Similar fits were achieved for all NP/cell combinations with the parameters described (refer to Supplementary Fig 10, results are presented as mean ± s.d. of $N = 3$ biological replicates, experiment was reproduced twice). (**b**) An association rate, or rate of uptake, was then derived from the data and normalized to the lowest rate (PLA NPs in neurons). Each point is the mean ± s.d. of $N = 3$ biological replicates. (**c,d**) *In vivo* MFI values for 4 and 24 h in healthy (**c**) and tumour-bearing (**d**) brains (in both cases, results are presented as mean ± s.d. of $N = 5$ biological replicates). (**e–h**) Linear regression fit of normalized rate of uptake *in vitro* to MFI values of specific cell populations *in vivo* (results are presented as mean ± s.d. of $N = 3$ biological replicates for the x axis and $N = 5$ biological replicates for the y axis). Linear regression analysis showed statistically significant slopes ($P < 0.001$) in all cases and a $R^2$ value ranging from 0.5387 to 0.8397. Each graph is followed by a corresponding schematic of different rate processes present in each condition, with rate of NP association with cells ($k_{on}$), rate of NP loss to capillaries ($k_{sys}$), rate of NP loss to CSF ($k_{CSF}$) and rate of cellular mitotic rate ($k_{mit}$) depicted with size of arrows.

different formulations, delivering the same dose of drug ($0.1\,mg\,kg^{-1}$) and the same number of NPs. The administration of free drug did not provide any significant improvement in survival compared to the control group (Fig. 8b and Supplementary Table 2), likely due to a rapid clearance of the drug from the extracellular space. On the other hand, administration of any of the NP formulations significantly increased the mean survival time (Fig. 8b and Supplementary Table 2). EB loaded PLA–PEG NPs provided the smallest increase in survival compared to the PBS-treated control group (21.5 and 16 days respectively), presumably due to the limited tumour cellular uptake (Fig. 6c,g) and the decreased drug release over the first days of treatment (Fig. 8a) that we identified with this material. On the other hand, EB loaded PLA and PLA–HPG–CHO NPs produced the longest survival times (33 and 28 days respectively), likely due to their more substantial internalization by tumour cells. An unexpected result was the extended survival of animals treated with EB loaded PLA–HPG NPs, which appeared to be as effective as the bio-adhesive formulation PLA–HPG–CHO NPs (mean survival times of 28 days for both formulations). This result, which did not correlate with our absolute internalization results, might be related to two features we identified in our studies for this NP formulation: PLA–HPG NPs were the only formulation that did not trigger an immune response (that is, activation of microglia) after infusion in the healthy brain (Fig. 4o), hence possibly avoiding clearance from the brain space, and PLA–HPG NPs demonstrated the highest specificity towards tumour cells 4 h

after administration to the tumour-bearing brain (Fig. 6d). It is also possible that for a very potent drug such as EB, the level of internalization and drug release of the PLA–HPG NPs was sufficient to reach the IC50 of the drug.

Taking a step further, we evaluated the short-term and long-term toxicity of the different formulations after CED in the healthy brain. Three days after infusion significant haemorrhage was noted in all groups (Fig. 8c). This was attributed to the trauma from the infusion catheter and disturbance of microvasculture, as the haemorrhage was localized to the catheter track. Within the volume where particles were distributed, significant amounts of monocyte infiltration were apparent for all particle types, especially circumscribing the large vessels within the brain. One noteworthy difference was observed in brains in which PLA–HPG–CHO NP was infused; in this case, a large density of cells, identified as neutrophils, were observed into the brain parenchyma (Fig. 8c-iv). Long-term toxicity was assessed by evaluating brains 3 weeks after NP administration (Fig. 8d). No significant changes were observed in the PLA–HPG and PLA–PEG NP infused brains. No signs of neurotoxicity was observed in the PLA and PLA–HPG–CHO NP infused brains, but hemosiderin-ladened macrophages were still present at this time point, clearing out the haemorrhage that was caused from the infusion, and perhaps pointing to delayed monocyte activity.

Cellular distribution of NPs within tissues is critically important for their therapeutic effectiveness, as demonstrated in recent studies on the cellular distribution of particles delivered to the liver[44] or tumours[45,46]. Here, we found that surface properties

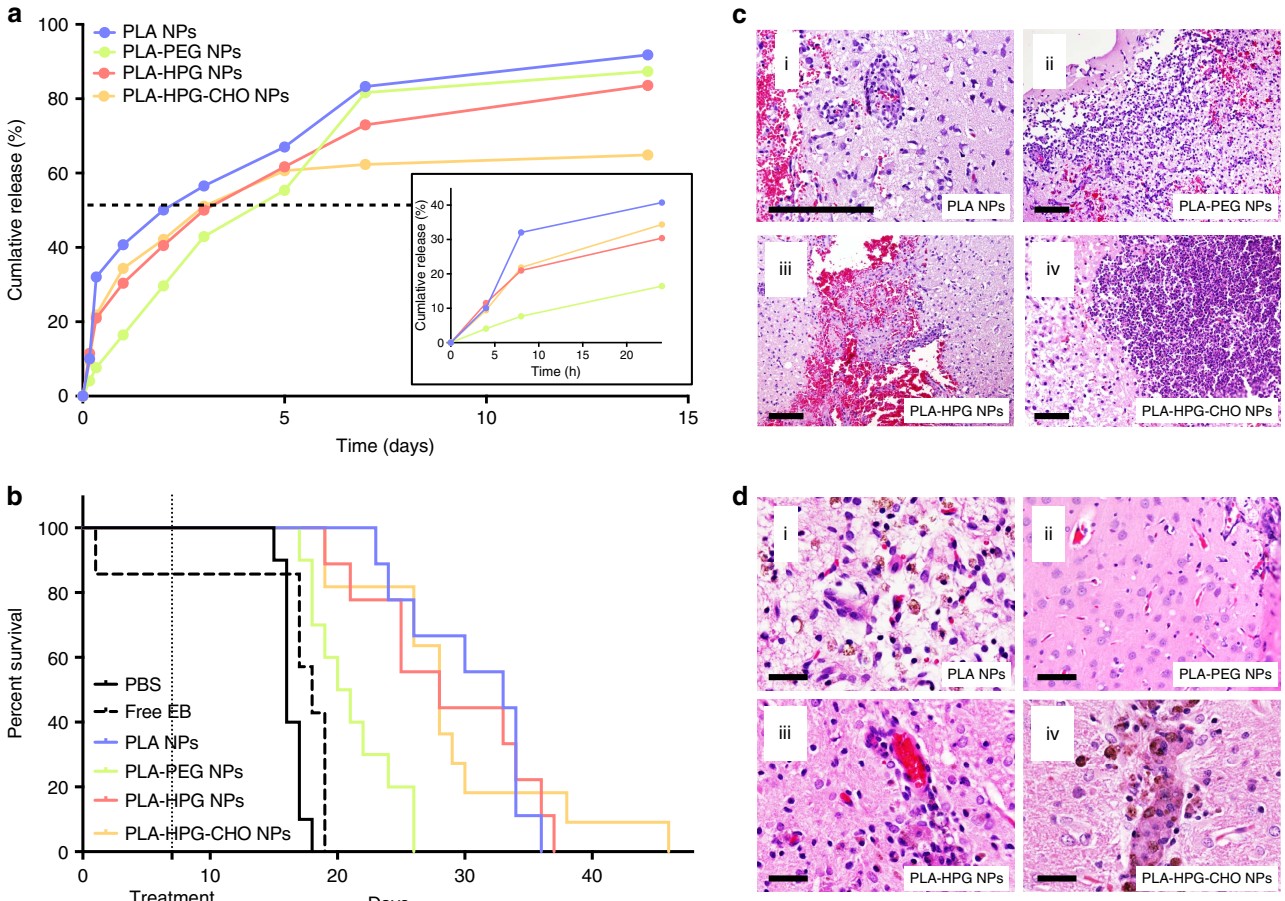

**Figure 8 | *In vivo* therapeutic efficacy and toxicity after administration by CED.** The four particle formulations were engineered to encapsulate the same amount of EB while retaining similar physico-chemical properties (Supplementary Table 2). (**a**) The four formulations presented the same release pattern, with a burst release during the first hours of incubation, followed by a slow release over the following days of incubation. This suggests that the overall drug release pattern is mainly governed by the nature of the NPs core (which is PLA for all the NPs formulation), and not dramatically influenced by surface modifications. (**b**) The four formulations were administered by CED to rats bearing RG2 tumours grown for 7 days to assess therapeutic efficacy. Administration of free EB did not significantly increased the mean survival time of the animals compared to the control treatment (PBS). In accordance with their internalization profiles, PLA and PLA–HPG–CHO NPs significantly extended the mean survival time, while the benefit observed after the administration PLA–PEG NPs was limited. Unexpectedly, PLA–HPG NPs provided the same survival benefit as the two most internalized formulations, despite limited uptake. (**c**) i–iv, haematoxylin and eosin (H&E) staining of brains, 3 days (short-term) after administration of the different formulation by CED. Microhemorrhages along the needle track from disturbance of microvasculture was noted in all groups, and in the PLA–HPG–CHO NP infused brains, large density of cells, identified as neutrophils, seemed to be recruited. (Scale bar, 200 μm) (**d**) i–iv, H&E staining of brains, 3 weeks (long-term) after administration of the different formulation by CED. No signs of neurotoxicity was observed in any group, but hemosiderin laden macrophages were still present at this time point in the PLA and PLA–HPG–CHO NP infused brains. (Scale bar, 50 μm).

of NPs are crucial in determining the cellular fate of NPs after entry into the brain interstitium. PEG and HPG moieties have been widely used to provide stealth properties in the systemic circulation[23], and dense PEG coating has been proposed as a means to enhance distribution in the brain interstitium[15], suggesting that stealth properties are also required for effective brain delivery. However, in our study, those particles were the least efficiently internalized by all cell types, in healthy brains and in tumour-bearing brains, and these particles were least effective for treatment of tumours (Fig. 8b). The lower internalization of stealth particles might be due to decreased retention in the brain environment resulting in reduced probability of interaction with cells and the extracellular matrix, making them more prone to elimination through brain capillaries[47] or the lymphatic system (Fig. 7). On the other hand, when the surface diols on PLA–HPG NPs were converted to aldehydes, the NPs were internalized more abundantly by all cell types. In this example, the multifunctional HPG was modified to produce a bio-adhesive state, which appears

to be a valuable strategy to increase cellular uptake after brain administration.

Our results also demonstrate that NP dynamics in the brain microenvironment can be modelled in cultured cells. This property may be specific to the brain environment, as introduction into the brain interstitium presents a situation in which fluid movements are slow compared to cellular uptake, which is recreated by static cell culture conditions. We hypothesize that the fate of a particle introduced into the brain interstitium is mainly determined by three rate-driven phenomena (Fig. 7e–h). The cellular association rate (or set of association rates, $k_{on}$) governs the likelihood that a NP will become associated with a cell. The NP clearance rate governs the rate of loss from the interstitial space due to transport from the interstitial fluid to the cerebrospinal fluid (CSF, $k_{CSF}$) or transport from the interstitial fluid to capillary blood ($k_{sys}$). Finally, the cell division rate ($k_{mit}$) determines the rate at which the number of cells able to internalize NPs increase, effectively changing the concentration of

particles that are available per cell. These three phenomena and their relative contributions to the fate of a NP population are dependent on the characteristics of the brain microenvironment: for example, in the healthy brain, the mitotic activity of the resident cells is very low, such that the third rate parameter may be negligible (Fig. 7e,f), whereas in the tumour brain, highly mitotic and metabolic tumour cells make this rate important (Fig. 7g,h). Our *in vivo* experimental data supports this model. Between 4 and 24 h after introduction of NPs into the brain interstitium, the increase of total uptake varied between particle types but the relative uptake by the different cell types remained unchanged for all formulations. This suggests that in an environment where cells have low mitotic or metabolic activity, surface properties of the NPs drive cellular association, which remains comparable over time. In tumour-bearing brains, mitotically active tumour cells, and their ability to activate microglia and to recruit TAMs, strongly influenced NP uptake depending on their surface properties. For example, NP association with tumour cells increased from 4 to 24 h by 34, 7, 5 and 11% for PLA, PLA–PEG, PLA–HPG and PLA–HPG–CHO NPs, respectively. For these intracranial tumours, the tumour cell content increased by 5% from day 7 to 8, suggesting that the increase in PLA–PEG and PLA–HPG NPs uptake was mainly due to cellular multiplication, while PLA and PLA–HPG–CHO NPs actively interacted with the highly metabolic tumour cells.

Our *in vitro* and *in vivo* results clearly demonstrate that NPs surface properties influence the amount of internalization by all cell types. Stealth formulations were shown to reduce cellular uptake both in the healthy brain and the tumour-bearing brain, while the introduction of bio-adhesive surface modifications dramatically enhanced the association of NPs with particular cell populations, such as tumour cells. In our survival study, these results correlated with therapeutic efficacy, except for the PLA–HPG NPs formulation that demonstrated an unexpected effectiveness in treating intracranial tumours when loaded with the chemotherapeutic drug EB. This observation suggests that the EB intracellular concentration reached with PLA–HPG NPs was sufficient to kill the tumour cells that internalized particles. Using EB loaded PLA–HPG–CHO NPs or PLA NPs to increase internalization did not provide further improvement in survival, and could even have been deleterious, as these NPs were shown to significantly disturb the brain microenvironment (Fig. 8c,d). It is important to note that in some cases, increased internalization does not always correlate with enhanced therapeutic efficacy, and can even lead to toxicity towards healthy tissue. Hence, evaluating the biocompatibility of nanoparticulate agents in parallel with their efficacy is of primary importance. In this particular case, improved therapeutic effect might be obtained not by increasing the absolute amount of drug internalized per cell, but by improving tumour coverage to ensure that all the tumour cells are internalizing particles[19]. On the other hand, PLA–HPG NPs performed better compared to the other stealth formulation, PLA–PEG NPs, in spite of similar level of internalization. This observation can be explained by two reasons: the PLA–HPG NPs do not trigger an immune response (that is, activation of microglia and/or recruitment of TAM; Fig. 4o), likely reducing clearance from the brain space compared to the immunogenic PLA–PEG NPs, and PLA–HPG NPs demonstrated higher specificity towards tumour cells (Fig. 6d) leading to a two-times higher absolute internalization by tumour cells at 4 h (Supplementary Table 1). This increase combined to a more effective drug release at short time points (Fig. 8a) might be sufficient to increase the intracellular drug concentration above its IC50. One should note that all those observations are likely to be specific to the use of a very potent chemotherapeutic drug such as EB, and that another therapeutic strategy might benefit from

the features of the other formulations. For example, a less potent drug might take advantage from a higher internalization provided by the PLA NPs or the bio-adhesive PLA–HPG–CHO NPs. Immunotherapy might also profit from the short-term neutrophil recruitment induced by the bio-adhesive PLA–HPG–CHO NPs (Fig. 8c), as it has been previously shown in conjunction with radiotherapy[48].

In conclusion, this study demonstrates that NP surface properties significantly influence cellular tropism after their entry into the brain, and that engineering these surface properties provides an opportunity to control cellular distribution. We have also demonstrated that *in vitro* rates of association can predict *in vivo* cellular affinity, showing that the dynamic and complex brain environment can be modelled, at least to some extent, by a simple static *in vitro* system. However, it appeared that the functional outcome might not always be related solely to an absolute internalization, and that the different features of the formulations (specificity of internalization, immunogenicity and ability to recruit specific cell types) should also be taken into account depending on the therapeutic strategy envisioned. Altogether, these results highlight the potential to optimize NP-based therapeutic strategies by tuning NP surface properties for enhanced delivery to certain cell types of interest and development of specific features, offering the possibility to hone the choice of the nanocarrier to each specific therapeutic molecule. Tailoring the nanocarrier behaviour to each therapeutic strategy will likely improve therapeutic efficacy in target cells, and minimize treatment toxicity to off-target cells.

## Methods

**Materials.** Poly(D,L-lactic acid; Mw = 20.2 kDa, Mn = 12.4 kDa) was purchased from Lactel. PEG-b-Polylactic acid di-block polymer (Mw PEG = 5 kDa, Mw PLA = 10 kDa) was purchased from Polysciences, Inc. (Warrington, PA, USA). Anhydrous N,N-dimethylformamide, dichloromethane, N,N′-diisopropylcarbodiimide, 4-(dimethylamino)pyridine, glycidol, potassium methoxide (25% in methanol), poly(vinyl alcohol) (Avg Mw = 30–70 kDa), paraformaldehyde, Tween 80 and 1,1,1-tris(hydroxymethyl)propane were obtained from Sigma-Aldrich (St Louis, MO, USA). Diethyl ether, ethyl acetate, methanol, acetonitrile and dimethylsulfoxide (DMSO) were obtained from J.T. Baker (Avantor Performance Materials, Central Valley, PA, USA). 0.5–1 kDa dialysis device (10 ml volume size) were purchased from VWR International (Bridgeport, NJ, USA).

*Hyperbranched polyglycerol Synthesis.* Hyperbranched polyglycerol (HPG) was synthesized as previously described through anionic polymerization[23]. 4.5 mmol 1,1,1-tris(hydroxymethyl)propane was added to a flask at 95 °C under Ar atmosphere, and 0.33 equivalent of potassium methoxide was added via syringe with stirring. The reaction was stirred for 30 min, and subsequently 25 ml of glycidol was added with a syringe pump over 12 h. The resulting product was dissolved in methanol and precipitated with acetone. After three rounds of precipitation, additional low molecular weight HPG was removed using dialysis (0.5–1 kDa MWCO) against deionized (DI) water over 24 h. HPG solution was collected, precipitated with acetone and dried under vacuum at 80 °C for 12 h.

*PLA–HPG synthesis.* PLA–HPG copolymer was synthesized as previously described[23]. PLA of 2 g and HPG at a 1:1 ratio were dissolved in 20 ml of dry N,N-dimethylformamide under molecular sieves overnight. 2 equivalents of N,N′-diisopropylcarbodiimide and 20 mol% of 4-(dimethylamino)pyridine were added and allowed to react for 5 days. The polymer was then precipitated with cold ether. The resulting precipitate was then re-dissolved in 10 ml of dichloromethane and precipitated with a mixture of cold ether and methanol and lyophilized for 2 days.

**Nanoparticle preparation.** *PLA NPs and PLA–HPG NPs.* PLA and PLA–HPG NPs were synthesized using the emulsion-evaporation technique, as previously described[23]. Polymer (100 mg) was dissolved in 2.4 ml of ethyl acetate overnight. 0.2 mg of DiA dye (Life Technologies, Grand Island, NY, USA) was dissolved in 600 µl DMSO and added to the polymer solution. The organic phase was then added dropwise to 4 ml of DI water under a strong vortex, and the mixture was sonicated for four cycles of 10 s intervals before dilution in another 10 ml of DI water. The final mixture was concentrated using a rotovap for 15 min at room temperature. Following evaporation, the solution was transferred to an Amicon Ultra-15 100 kDa centrifugal filter unit, and centrifuged at 3,000g at 4 °C for 45 min. The NPs were washed twice with 15 ml DI water and centrifuged for 45 min each time. Subsequently, the NPs were re-suspended in 1 ml DI water, and snap-frozen in aliquots until use.

*PLA–PEG NPs.* PLA–PEG NPs were synthesized using a nanoprecipitation technique. Polymer (100 mg) was dissolved in 5 ml DMSO at room temperature for 2 h. DiA dye (0.2 mg) dissolved in 2 μl of DMSO was then added to the polymer solution. The polymer-dye solution was then divided into 200 μl aliquots. Each 200 μl aliquot was added dropwise to 1 ml DI water under strong vortex to create a NP suspension. These suspensions were immediately pooled and diluted with 5 × DI water. This diluted suspension was then transferred to an Amicon Ultra-15 100 kDa centrifugal filter unit, and centrifuged at 4,000*g* at 4 °C for 30 min. The NPs were washed twice with DI water and centrifuged for another 30 min each time. After a final wash with DI water, the NPs were centrifuged for 1 h to achieve a final concentration of 100 mg ml$^{-1}$ DI water. The final NP suspension was then either immediately used for *in vivo* or *in vitro* experiments, or snap-frozen at −80 °C until use.

*PLA–HPG–CHO NPs.* PLA–HPG–CHO NPs were prepared by oxidation of the vicinal diols of PLA–HPG NPs to aldehydes as previously described with minor modifications[24]. PLA–HPG NPs (200 μl) at 100 mg ml$^{-1}$ were incubated for 20 min on ice with 60 μl of 10 × PBS and 200 μl of 0.1 M NaIO$_4$(aq). The reaction was then quenched with 200 μl of 0.2 M Na$_2$SO$_3$(aq) and the NPs were washed two times with DI water using Amicon Ultracel 100 kDa MWCO centrifugal filter units at 12,200*g* for 7 min each. The NPs were then diluted with DI water to desired concentrations for experiments.

*EB loaded NPs.* EB loaded NPs were prepared similar to dye loaded particles. Instead of 0.2 mg of DiA dye, 2.5, 5 and 15 mg of EB was used per 100 mg of PLA, PLA–HPG and PLA–PEG polymer respectively.

**Nanoparticle characterization.** *Physico-chemical properties of nanoparticles.* The hydrodynamic diameter of the NPs was measured by dynamic light scattering. NPs (1 ml; 0.05 mg ml$^{-1}$ in DI water) was prepared and read on a Malvern Nano-ZS (Malvern Instruments, UK). To measure zeta potential, 750 μl of NPs (0.05 mg ml$^{-1}$ in DI water) were loaded into a disposable capillary cell and analysed on a Malvern Nano-ZS. For TEM imaging, 10 μl of NP solution at a concentration of 10 mg ml$^{-1}$ was placed on a pre-cleaned and hydrophilized CF400-CU TEM grid (Electron Microscopy Sciences, Hatfield, PA, USA) for 1 min. Grids were stained with a 0.2% uranyl acetate solution for 15 s, washed three times in DI water and mounted for imaging with a Tecnai T12 TEM microscope (FEI, Hillsboro, OR, USA). Particle stability was measured using Malvern Nano-ZS in artificial cerebrospinal fluid (Harvard Apparatus, Holliston, MA, USA) at 37 °C with a standard operating procedure taking measurements every minute.

*Particle yield and dye loading.* A 100 μl solution of NPs was lyophilized in a pre-weighed eppendorf tube to measure particle yield. Dye loading was determined using a SpectraMax M5 plate reader (Molecular Devices, Sunnyvale, CA, USA) at 456/590 (nm). EB loading was calculated by dissolving a 10 μl aliquot of the NPs in 90 μl of Acetonitrile for 24 h at 37 °C. Solutions were then ran through a Agilent LC-MS 6120B (Agilent Technologies, Santa Clara, CA, USA) with standard curves to determine the loading of drug in NPs. The release of EB from the different NPs formulations was measured for up to 14 days. NPs loaded with 2% wt/wt drug were dispersed at 0.5 mg ml$^{-1}$ in 1 × PBS with 0.5% tween 80, and incubated at 37 °C. At different time points (4 h, 8 h, 24 h, 48 h, 72 h, 5 days, 7 days and 14 days), the suspension was centrifuged through a 3 kDa filter. Filtrate was collected for HPLC analysis as previously described for EB loading, and the pellet was re-suspended in the same volume of PBS for continued release.

**Cellular tropism of NPs following CED.** *Experiment rationale.* For volume of distribution experiments, a biological replicate of three was performed to ensure reproducibility and accuracy of our data, as previously reported in literature by our group. For flow cytometry, a biological replicate of five was performed to repeat the experiment on three different days, accounting for any processing variables and to ensure day to day variation. Moreover, preliminary experiments demonstrated that $N = 5$ was sufficient to ensure the detection of significant differences. Animals that died during procedures were not processed further for analysis. For the therapeutic efficacy study, an initial number of 10 animals per group was previously shown in our group to ensure statistical differences. Animals dying without waking up from surgery (after tumour implantation or CED) were excluded. Animals were randomly assigned to the different groups to prevent any confounding variables from syringes or stereotactic frames. Investigators were not formally blinded to experimental groups.

All procedures were performed in accordance with the guidelines and policies of the Yale Animal Resource Center (YARC) and approved by the Institutional Animal Care and Use Committee (IACUC). Male Fischer 344 rats (Charles River Laboratories, 200–220 g) were used. Surgical procedures were performed using standard sterile surgical techniques.

*Convection-enhanced delivery in the healthy brain.* Animals were anaesthetized using a mixture of ketamine (75 mg kg$^{-1}$) and xylazine (5 mg kg$^{-1}$), injected intraperitoneally. Rats' heads were shaved and then placed in a stereotaxic frame. After sterilization of the scalp with alcohol and betadine, a midline scalp incision was made to expose the coronal and sagittal sutures, and a burr hole was drilled 3 mm lateral to the sagittal suture and 0.5 mm anterior to the bregma. A 50 μl Hamilton syringe with a polyamide-tipped tubing, loaded with the NPs, was inserted into the burr hole at a depth of 5 mm from the surface of the brain and left to equilibrate for 7 min before infusion. A micro-infusion pump (World Precision

Instruments, Sarasota, FL, USA) was used to infuse 20 μl of 50 mg ml$^{-1}$ NPs at a rate of 0.667 μl min$^{-1}$. Once the infusion was finished, the syringe was left in place for another 7 min before removal of the syringe. Bone wax was used to fill the burr hole and skin was stapled and cleaned. Following intramuscular administration of analgesic (Meloxicam, 1 mg kg$^{-1}$), animals were placed in a heated cage until full recovery.

*Orthotopic tumour inoculation.* Orthotopic RG2 tumours were inoculated as previously described. Briefly, 3 μl containing $2.5 \times 10^5$ RG2 cells suspended in PBS were administered over 3 min using the same procedure and the same coordinates as CED. Tumours were grown for 7 days before administration of particles.

*Convection-enhanced delivery in the tumour-bearing brain.* CED in tumour-bearing rats was conducted following the exact same procedure as for the healthy rats, by reopening the burr hole used for tumour implantation.

*Volume of distribution.* Brain was harvested immediately after infusion and flash frozen. Brain was sliced in 50 μm slices using a Leica Cryostat CM3000 (Leica, Germany). Slides were imaged using a Zeiss Lumar.V12 stereoscope (Carl Zeiss AG, Germany) and images were analysed using a MATLAB code setting a threshold with Otsu's method.

*Flow cytometry.* Animals were euthanized 4 or 24 h after CED and brains were harvested. The olfactory bulb, cerebellum and contralateral hemisphere to the injection site were removed. A similar procedure was completed with isolation of the striatum, but yielded similar results; therefore the entire hemisphere was used to increase the number of counts per sample. The following method has been optimized from studies previously described and validated[35,49,50]. The brain was diced into small pieces and suspended in 4 ml of PBS. Tissue was further broken down by serial pipetting and diluted to 25 ml until it passed through a 40 μm cell strainer with no resistance. After pelleting the cells, 3 ml of ammonium chloride potassium (ACK) lysing buffer (Lonza, Switzerland) was added to the suspension to lyse red blood cells. PBS of 22 ml was immediately added to neutralize the ammonium chloride potassium (ACK) buffer, and the cell suspension was then centrifuged at 950*g* for 10 min and re-suspended in 4 ml of PBS. Cell suspension of 2 ml was gently placed on top of a 4 ml 20% Percoll solution (GE Healthcare Bio-Sciences, Pittsburgh, PA, USA) in DMEM/F-12 and centrifuged at 3,000*g* for 15 min to separate out a lipid/myelin layer. This layer was removed and the rest of the solution was diluted 25 × and then centrifuged for 10 min at 1,000*g* to collect the cells. The cells were re-suspended in 2 ml of PBS. A small aliquot was dyed with Trypan Blue and counted for cell density and viability before proceeding with staining procedures. Two millilitre of 8% PFA solution was added to fix the cells for 10 min. After 2 washing steps (1,000*g* for 10 min), cells were re-suspended in 3 ml of 0.1% triton-X solution to permeabilize for 10 min. After 2 washing steps (1,000*g* for 10 min), non-specific binding of antibodies was prevented by blocking for 30 min in a 3 ml 10% BSA solution, followed by incubation with conjugated primary antibodies (NeuN/FOX3-Cy5 (1613R), GFAP-Cy5 (0199R), AIF-1/Iba1-Cy5 (1363R); Bioss Antibodies, Woburn, MA, USA; GFP-488, Life Technologies) at a concentration of 1 μg per 250 000 cells for 30 min. Finally, cells were washed three times with 1 ml of a 10% BSA solution for 10 min and re-suspended in 320 μl of a 1% BSA solution for flow cytometry. Flow cytometry was performed using an Attune NxT (Invitrogen, Carlsbad, CA, USA) with the flowing laser voltages; FSC, SSC, GFP (BL1), DiA (BL2), Cy5 (RL1): 640, 420, 400, 400, 500, respectively, and 300 000 iterations were acquired. Experiments for several animals of a single particle type were done on different days to ensure reproducibility and to account for any random experimental biases.

*Flow cytometry analysis.* Data were analysed using FlowJo v.10.0.8r1 (FlowJo, Ashland, OR, USA). After selecting for viable mononuclear cells under FSC and SSC, a population shift in the Cy5 channel for each sample allowed for gating specific cell populations (refer to Supplementary Fig. 1b, adapted from Ngambenjawong et al.[45]). The fluorescence histograms of the cells gated in the DiA channel were then superimposed with the histogram of the control brain cell populations, to gate the cell populations that shifted out of the control histogram. The MFI in the DiA channel of this final population was then recorded. MFI values were then normalized according to dye loading using the relative fluorescence of each NPs preparation obtained through a plate reader measurement. Since MFI is the measurement of the fluorescence intensity of a cell population, or $\frac{\text{Absolute Fluorescence}}{\text{\# of cells in population}}$, to deduce what the absolute fluorescence within a cell population was, the MFI had to be multiplied by the relative percentage of the population of cells to yield the absolute fluorescence internalized by the cells:

$$\frac{\text{Absolute Fluorescence}}{\text{\# of cells in population}} (\text{or MFI}) \times \frac{\text{\# of cells in population}}{\text{Total \# of cells}} (\text{or \% of cells in population}). \tag{1}$$

Since the total # of cells for each sample was set to a fixed amount during our flow cytometry measurements, MFI × % of cells in population was proportional to the Absolute Fluorescence and could be compared between animals and samples. Finally, the sum of these fluorescence intensities within each cell population yielded the total fluorescence, or particles uptake, and was depicted through the size of the pie charts in the figures. A normalized value of particle amount in each cell type is presented in Supplementary Table 1. The per cent cell shift outside the control population was calculated by creating a quadrant gate in the DiA channel for the

control populations. The values in Fig. 3b, Supplementary Figs 5 and 12 are an average of $N = 4$–6.

**Immunostaining and imaging.** For immunostaining and imaging, brains were fixed in 4% PFA for 24 h and placed in a 30% sucrose solution until it equilibrated and sank. Tissue was then sliced using a Leica CM3000 Cryostat (Leica) to 10 μm slices and stored at −20 °C until staining. Just before staining, slides were rehydrated with PBS and then permeabilized with 0.1% triton-X for 1 h. The slides were then washed three times and incubated in blocking buffer (5% BSA, 5% donkey serum and 0.05% triton-X solution) for 1 h. Slides were incubated with primary antibodies (Anti Iba1 Rabbit; WAKO Pure Chemicals, Richmond, VA, USA; Anti GFAP Rabbit, bs-0199R; Bioss Antibodies; Anti NeuN Rabbit, ab1044225; Abcam, Cambridge, MA, USA) diluted to 1:500 in blocking buffer for 18 h at 4 °C. Slides were washed three times with blocking buffer for 10 min each and then incubated with secondary antibodies (Goat anti Rabbit Cy5, A10523; Life Technologies) diluted to 1:1,000 in blocking buffer for 1 h at room temperature. Finally, slides were washed two times with blocking for 10 min and once with water, then mounted for imaging using VECTASHIELD HardSet Antifade Mounting Medium with DAPI (Vector Laboratories, Burlingame, CA, USA). Images were taken using a Leica TCS SP5 confocal microscope (Leica).

**Therapeutic efficacy study.** The therapeutic efficacy study was conducted following the exact same procedure as for the CED of dye loaded particles, using EB loaded particles. Each rat received a total dose of 20 μg of EB.

**Toxicity study.** To evaluate the toxicity of the different formulations, healthy rats were infused with the four different particles type loaded with the DiA dye. Three days (short-term toxicity) or 3 weeks (long-term toxicity) after the infusion, the rats were killed, the brains were harvested and fixed in 4% PFA for 24 h before haematoxylin and eosin staining. A certified neuropathologist blindly evaluated and scored the slices.

**Cell culture.** N27 (rat neural cell) were obtained from EMD Millipore (Billercia, MA, USA) and DI TNC1 (rat astrocyte) and RG2 (rat glioma) cell lines were obtained from ATCC (Manassas, VA, USA). BV-2 (murine microglia) cells were a kind gift from Dr Hideyuki Takahashi from Yale Cellular Neuroscience department. N27 cells were cultured in RPMI 1640 media supplemented with 10% FBS and 1% pen/strep. DI TNC1, RG2 and BV-2 cell lines were cultured in 4.5 g l$^{-1}$ glucose DMEM media supplemented with 10% FBS and 1% pen/strep.

**Uptake kinetic studies.** Cells were plated at a density of 10,000 cells per well in 96-well plates. 24 h after, cells were treated with fluorescent particles at a concentration of 1 mg ml$^{-1}$. Cells were incubated with particles for different time points (30 min, 1 h, 2 h, 4 h, 6 h, 8 h, 12 h and 24 h) washed thoroughly three times with a warm 1% BSA solution before adding trypsin and re-suspending cells in a cold 1% BSA solution on ice. Flow cytometry was performed using Attune NxT (Invitrogen) and at least 5,000 iterations were acquired, then the data were analysed using FlowJo v.10.0.8r1. Plots were fitted with the association kinetics equation on Prism 6 with the restraints assumption of dissociation rate being 0. This was done because we assumed Koff (off rate of particles) was negligible in our *in vitro* system, allowing the direct comparison of the association rate as a predictive value of *in vivo* outcomes. The parameter of Hotnm (particle concentration) was not significant, because it cancelled out as a normalizing factor and allowed us to derive a normalized rate of uptake of particles by cells independent of the maximum intensity. Linear regression of relative rate of uptake *in vitro* versus MFI values *in vivo* was also performed in Prism.

**Graphing and analysis.** Prism 6 was used for graphing and statistical analysis. Statistical significance was tested using a two-tailed Student's *t*-test, two-way analysis of variance (Supplementary Data 1) and linear regression was analysed using Prism 6's analysis software. Flow cytometry data analysis was done with FlowJo v.10.0.8r1 and Microsoft Office Excel 2011. Image analysis was done with MATLAB R2015a and ImageJ (FIJI plugin). Image processing was done using LAS AF (Leica) and graphical schematics were made on Adobe Photoshop CS6.

**Data availability.** Contact corresponding author for MATLAB code regarding analysis of volume of distribution of NPs in the brain.

Data supporting the findings of this study are available within the article and its Supplementary Information Files and from the corresponding author upon reasonable request.

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

## Acknowledgements

This work was supported by the National Institute of Health (NIH R01 CA149128) and the Brain Research Foundation. A.G. was supported by the Leslie H. Warner postdoctoral fellowship from the Yale Cancer Center. The authors thank Marc Llaguno for TEM assistance. We thank Muneeb Mohideen, Elias Quijano, Junwei Zhang, David Chi, Dr Fred Gorelick and Dr Jiangbing Zhou for insightful discussions and Paul Won for technical assistance.

## Author contributions

E.S., A.G. and W.M.S. conceived and designed the research. H.-W.S and Y.D. synthesized and characterized polymers used in this paper. A.G., A.R.K. and E.S. designed and performed the NP preparation and subsequent in vitro and in vivo characterizations. A.G., Y.-E.S. and E.S. performed all animal surgeries. E.S. and A.G. designed in vivo cellular tropism experiments and E.S. executed the experiments. E.S. and J.C. performed staining procedures and subsequent confocal imaging. G.T.T., E.S. and A.G. designed the in vitro kinetic experiments, E.S. conducted the experiments and A.G., E.S. and G.T.T. analysed the data. A.H. analysed histological data. E.S. and A.G. analysed all other data. A.G., E.S. and W.M.S. co-wrote the paper. All authors discussed the results and commented on the manuscript.

## Additional information

**Competing interests:** The authors declare no competing financial interests.

