## [Peer Review File · Nature Communications]

Reviewers' comments:

Reviewer #1 (Remarks to the Author):

The manuscript examines the surface properties of nanoparticles that influence retention in the various cells within the brain following CED and the impact of surface properties on drug response in both in vitro and in vivo models of brain tumor.

The main hypothesis being that nanoparticle diffusion away from the infusion site is governed by how effectively it binds to and is internalized by the cells within the brain. The negatively charged PLA nanoparticles were coated with stealth polymer PEG and hyperbranched glycerol (HPG) and nanoparticles with bioadhesive properties. Such a study with highly controlled size and charge properties is important and necessary to truly evaluate the extent of the polymer surface properties. The techniques used to map NP uptake in cells both in culture and in vivo, using FAC and dual fluorescence is highly advanced.

That being said, I have the following concerns:

1. The investigators report a high amount of uptake of the NPs into the brain cells following CED, however, it would be very important to know of the fluorescence activity administered into the brain using CED, how much of the fluorescence is associated with the cells. Currently the data is presented as a breakdown of NP uptake into the various cells with neurons showing the least and the microglia and tumor showing the most. But how much of the nanoparticles injected into the brain does this represent. It is important as it gets at the efficiency of the delivery. Investigators rightfully point out about 1% or less of administered dose is delivered to the brain, what fraction of the total NPs infused into the brain is captured in the fluorescence observed in the various cells? It would seem that the efficiency of uptake is different in the various NP formulations, to what extent do the stealth particles simply reside along the extracellular infusion path and/or removed through CSF?
2. The assumption that fluorescence intensity increases over time from 4 hours to 24 hours is that this reflects increased internalization of the NPs into the various cells, mostly microglia and tumor. However, to what extent is this truly internalized NPs. Any additional evidence as to the internalization of the NPs. Reference is made to potential routes of internalization. What evidence does the investigator have that the NPs that are captured in the fluorescence images is really internalized. While there is some overlap in the dual dye labels there are large areas that look to have fluorescence of the NPs and the various marker molecules that do not co-localize. Any evidence from either the in vitro or in vivo studies that the NPs do internalize to a greater extent at 24 vs 4 hrs.
3. The tumor responsiveness to the drug when given by PLA alone seems just as effective as the other more stealth NPs. There are clearly differences in the PLA vs stealth NPs in terms of cell uptake. This seems to be contradicted in the tumor response studies. Potential differences in the cellular processing of the NPs or drug release from the NPs could certainly be involved. To what extent is drug release similar or divergent in the various NP formulations? Furthermore, does the NP even have to be internalized to be effective if it resides within the extracellular spaces releasing drug.

Reviewer #2 (Remarks to the Author):

This paper explores some very important topics that are the distribution of nanoparticles in the brain depending on surface chemistry. The biomedical need behind is the need to cross the blood-brain barrier, that is to date possible with only a very poor efficacy using most nano-platforms developed. Investigation of convection enhanced delivery is a very important strategy, as this technology is now translated at the bedside, demonstrating efficacy in glioblastoma. The determination of the

cell type specificity is also very important as the brain function is mainly governed by network and very specific cell types. Transfecting in the same location a different cell type can induce opposite effects.

The nanoparticles used in this paper are PLA based nanoparticles and the mode of delivery is convection enhanced delivery. The main objective of this paper was to explore the transfection efficacy in connexion with surface modifications such as « stealth » properties and bio-adhesive modification. PLA, PLA PEG, PLA-HPG (avoiding phagocytic recognition) and PLA-HPG-CHO (bio-adhesive) were investigated. Moreover, an innovative in vitro cell culture test was implemented to predict transfection efficiency. The quality of the experimental data is high with excellent statistical analysis.

The main claim of this paper is that surface nanoparticles properties influence transfection efficacy as well as the cellular tropism of transfection and that it can be predict in vitro using the innovative test developed in this paper. Interestingly, stealth formulation decreases cellular uptake, while bio-adhesive surface modifications dramatically enhance cellular uptake. Correlated efficacy was demonstrated in a glioblastoma model, especially for PLA-HPG and PLA-HPG-CHO nanoparticles that are similarly efficient. Indeed, a low biocompatibility is observed for the adhesive nanoparticles, that induce a strong gliosis as well as a microglial activation and neutrophils infiltration.

In conclusion, this is an important work demonstrating for the first time relationship between surface properties of nanoparticles and transfection as well as cellular distribution inside the brain after convective infusion. The low impact of stealth modification for local delivery was not anticipated. The impact of adhesive modifications is new. A therapeutic efficacy is also demonstrated. An important drawback in this study is the toxicity observed with the more efficient and innovative nanoparticle harboring adhesive properties inducing an inflammatory gliotic reaction.

Reviewer #3 (Remarks to the Author):

This work investigates the cellular fate of PLA-based nanoparticles with various surface chemistries injected into the brain. Internalization is enhanced with particles coated with HPG with aldehyde functionality, as also shown in their in vitro studies. These particles were used to delivery epothilone B in an orthotopic model of glioblastoma.

The work is quite thorough and includes particle characterization, distribution analysis after injection into the brain of both normal and tumor bearing mice by imaging and flow cytometry, and in vivo efficacy using these nanoparticles to deliver drug to a glioblastoma model. Overall, I am quite impressed with the amount of work covered by the article.

However, my major reservation is that there is not much new information provided to the scientific community from the studies. Much of their findings have been previously reported in various publications (although this study is certainly the most comprehensive compilation). The impact of the work is therefore not very high. Another concern is that some of the graphs particularly the pie charts misrepresent the data.

Some more specific comments are below:

1. The DiA could significantly alter the physicochemical characteristics of the particles. This dye also inserts into cell membranes and is used as cell tracking studies, which could further confound the studies.
2. Flow cytometry of brain homogenates is challenging and methods used need to be more thoroughly outline and validated, including actual dot plots from flow cytometry and details of methods such as removal of fat.
3. Figure 4 states that HPG and HPG-CHO particles do not ramify microglia but it looks like all microglia in these figures are ramified. Normal untreated brain needs to be shown and GFAP-stained images moved to main text.
4. Figure 5: Glial population increases from day 7 to 8 and its not clear if this is due to nanoparticle treatment or tumor growth. Controls should be included.

5. In Figure 6, it appears the authors stained for Iba1, GFAP, GFP, NeuN and then gated each cell marker based off DiA. Then took the number of positive DiA cells and the relative MFI for those positive cells and multiplied by the number of Iba1 (or GFAP/NeuN or GFP) positive cells. Instead for readers it would be helpful to first gate off total cell number and tell the reader the total number of positive DiA cells within that total number. Then break down what percentage of that total DiA positive cells are Iba1, GFAP, GFP and NeuN positive. Level of brightness can be shown in supplemental information.

Aswers to reviewers' comments:

Reviewer #1 (Remarks to the Author):

The manuscript examines the surface properties of nanoparticles that influence retention in the various cells within the brain following CED and the impact of surface properties on drug response in both in vitro and in vivo models of brain tumor.

The main hypothesis being that nanoparticle diffusion away from the infusion site is governed by how effectively it binds to and is internalized by the cells within the brain. The negatively charged PLA nanoparticles were coated with stealth polymer PEG and hyperbranched glycerol (HPG) and nanoparticles with bioadhesive properties. Such a study with highly controlled size and charge properties is **important** and **necessary** to truly evaluate the extent of the polymer surface properties. The techniques used to map NP uptake in cells both in culture and in vivo, using FACS and dual fluorescence is **highly advanced**.

That being said, I have the following concerns:

1. The investigators report a high amount of uptake of the NPs into the brain cells following CED, however, it would be very important to know of the fluorescence activity administered into the brain using CED, **how much of the fluorescence is associated with the cells**. Currently the data is presented as a breakdown of NP uptake into the various cells with neurons showing the least and the microglia and tumor showing the most. But **how much of the nanoparticles injected into the brain does this represent**. It is important as it gets at the efficiency of the delivery. Investigators rightfully point out about 1% or less of administered dose is delivered to the brain, what fraction of the total NPs infused into the brain is captured in the fluorescence observed in the various cells? It would seem that the efficiency of uptake is different in the various NP formulations, **to what extent do the stealth particles simply reside along the extracellular infusion path and/or removed through CSF?**

We would like to thank the reviewer for his/her kind evaluation of our work and thoughtful questions.

Here, the reviewer raises a concern regarding our ability to quantify how much of the injected NPs is actually delivered to the brain compartment. Using Convection-Enhanced Delivery (CED) theoretically ensures the delivery of 100% of the dose, as this technique involves the direct infusion in the brain parenchyma, circumventing the limitations of the BBB. Our CED method has been optimized in terms of volume delivered, flow rate and needle design to prevent any backflow during the infusion, thus ensuring the delivery of 100% of the dose to the brain parenchyma. However, after the end of the infusion, we agree that NPs could be residing in the extracellular infusion path, and might eventually be removed through the brain vasculature to reach the systemic circulation or through the CSF, as suggested by the reviewer, leading to a decrease of the effective dose present in the brain compartment. We have thought about the importance of measuring the presence of NPs in both the CSF and the blood compartment. Unfortunately, several technical issues arose when trying to evaluate this. Especially, when attempting to measure the amount of NPs that escaped into the systemic circulation or CSF, the dilution of the particles into the fluid compartments made it undetectable with our current methodologies. Even after increasing the dye loading by 10 fold, the particles were still not detectable. If anything, these results suggested that the amount of NPs in these compartments is very low, and that a more sensitive methodology should be developed, which we think is out of the scope of this study. **Overall, the loss of particles after administration by CED seems to be minimal and we confidently consider that the fluorescence we are detecting represents 100% of the delivered dose.**

Another concern of the reviewer is to know how much fluorescence is actually associated with the cells, and how much resides in the extracellular space of the brain. The processing of the tissue in order to get a FACS-appropriate cell suspension results in the loss of NPs that are not internalized, thus not allowing us to accurately measure the amount of NPs in the extracellular space compartments using the FACS method. However, we still agree with the reviewers that this is an important question that needs to be addressed, and so here we developed another method for measuring the amount of NPs internalized versus the amount of NPs in the extracellular space by

using confocal imaging: we isolated sites of perinuclear uptake of particles and compared the fluorescence intensity at those sites to the fluorescence intensity in the rest of the image (Fig. 1).

Fig 1: Quantification of perinuclear signal in confocal images. Confocal images were analyzed using ImageJ. Cell nuclei were located in the DAPI channel, and circles with a diameter of 15 μm were drawn around each nuclei. Circles with MFI values higher than background in the DiA channel were selected as 'cells with perinuclear uptake', and the MFI in those circles was extracted. The % perinuclear MFI was calculated as the ratio of the MFI of 'cells with perinuclear uptake' over the total MFI of the image.

When comparing the 4 h images to the 24 h images, we observed a drastic difference in the measurement of the perinuclear population of NPs. For all NPs types and condition, the percentage of perinuclear NPs increased between 4 h and 24 h, demonstrating the internalization of the NPs. In particular, for the PLA-HPG-CHO NPs, the perinuclear NPs accounted for up to 80% of the fluorescent signal. We would like to thank the reviewer for this valuable suggestion, which we believe has increased the significance of our results. **These new results have been added to the manuscript as Supp. Fig 8.**

2. The assumption that fluorescence intensity increases over time from 4 hours to 24 hours is that this reflects increased internalization of the NPs into the various cells, mostly microglia and tumor. However, to what extent is this truly internalized NPs. Any additional evidence as to the internalization of the NPs. Reference is made to potential routes of internalization. **What evidence does the investigator have that the NPs that are captured in the fluorescence images is really internalized.** While there is some overlap in the dual dye labels there are large areas that look to have fluorescence of the NPs and the various marker molecules that do not co-localize. **Any evidence from either the in vitro or in vivo studies that the NPs do internalize to a greater extent at 24 vs 4 hrs.**

Here, the reviewer is asking about our ability to decipher between an internalization of NPs compared to their association with cell membranes that do not necessarily translate into actual uptake. This is an important point that is often discussed in the field of NPs based drug delivery. Following our new analysis of our confocal images (Fig 1 above), we feel confident that all NPs types are internalized as they end up mostly in the perinuclear space, which is clearly identifiable on confocal images. In order to support those results and show that our NPs were indeed internalized and not associated with the membrane, we performed an in vitro uptake experiment, using representative cell lines of the different

cell types found in the brain. After incubating the different cell lines with particles either at 37°C or at 4°C, we measured the NPs uptake by flow cytometry (Fig. 2).

Fig. 2: In vitro cellular uptake at 37°C and 4°C. The four formulations were incubated with representative cell lines of the different cell types found in the brain (TNC-1 cells for astrocytes, BV-2 cells for microglia and N27 cells for neurons), for 4 h at 37°C (plain bars) or 4°C (striped bars). Cellular internalization was assessed by flow cytometry. For all cell types and all formulations, the uptake at 4°C did not exceed 10% of the uptake at 37°C, confirming an active internalization of all NPs formulation by the brain cells.

We observed that when the cells were incubated at 4°C in presence of the different particle formulations, the level of internalization never exceeded 10% of the level reached when the incubation was performed at 37°C. Together with the results previously presented in Fig 1, these data demonstrate that the NPs are indeed internalized, via an energy-dependent uptake mechanism. **Once again, we would like to thank the reviewer for this valuable comment; these new results have been added to the manuscript as Supp. Fig 9.**

Regarding our confocal images, it is true that there are some areas with limited overlap between the various cell markers and NP fluorescence. However, we performed our image analysis with the following criteria in mind: NPs were considered internalized by a cell, if the fluorescence was localized in the perinuclear space of a cell stained by a cellular marker. Indeed, the cell markers that we used stain for different cellular compartments (IBA stains a surface protein, GFAP stains an actin protein, NeuN stains a nuclear protein) do not necessarily reside at the perinuclear site where the NPs may reside.

3. The tumor responsiveness to the drug when given by PLA alone seems just as effective as the other more stealth NPs. There are clearly differences in the PLA vs stealth NPs in terms of cell uptake. This seems to be contradicted in the tumor response studies. Potential differences in the cellular processing of the NPs or drug release from the NPs could certainly be involved. **To what extents is drug release similar or divergent in the various NP formulations? Furthermore, does the NP even have to be internalized to be effective if it resides within the extracellular spaces releasing drug.**

We agree with the reviewer that the efficacy results were surprising in relation to the uptake studies. As stated in our discussion, an unexpected result was the extended survival of animals treated with EB loaded PLA-HPG NPs, which appeared to be as effective as the bioadhesive formulation PLA-HPG-CHO NPs and the non-modified PLA NPs, despite the differences in cell uptake. The reviewer's

question regarding the possibility of different drug release profiles is an excellent suggestion, and we performed a drug release study to evaluate if there were any differences amongst the different formulations (Fig. 3).

Fig 3: Etoposide B (EB) cumulative release from the different NPs formulations after incubation in PBS at 37°C. The release of EB from the different NPs formulations was measured for 14 days. NPs loaded with 2% wt/wt drug were dispersed at 0.5 mg/mL in 1X PBS with 0.5% tween 80, and incubated at 37°C. At different time points (4 h, 8 h, 24 h, 48 h, 72 h, 5 days, 7 days, 14 days), the suspension was centrifuged through a 3 kDa filter. Filtrate was collected for HPLC analysis and the pellet was resuspended in the same volume of PBS for continued release. Insert shows a zoom in on the first 24 h of release.

We observed that the four formulations presented the same release pattern, with a modest burst release during the first hours of incubation, followed by a slow release over the following days of incubation. This suggests that the overall drug release pattern is mainly governed by the nature of the NPs core (which is PLA for all the NPs formulation), and not dramatically influenced by surface modifications. However, after 24 h (see insert), the PLA NPs had released 40% of their payload, the PLA-HPG and PLA-HPG-CHO NPs around 30%, and PLA-PEG NPs only 16%, suggesting that surface modifications can influence the burst release phase to some extent. Similarly, after 14 days, the PLA, PLA-HPG and PLA-PEG formulations released up to 80 to 90% of the drug, while the PLA-HPG-CHO NPs formulation released only 65%. The initial decreased release of drug from the PLA-PEG NPs is in accordance with a decreased survival efficacy. As stated by the reviewer, the PLA-HPG NPs formulation was as efficient as the unmodified PLA NPs, although they were (1) releasing less drug over the first 24 h and (2) being significantly less internalized by all cell types. It is possible that, for a very potent drug such as EB, the level of internalization and drug release of the PLA-HPG NPs was sufficient to reach the IC₅₀ of the drug. In that case, more internalization and more release obtained with the PLA NPs did not add any additional therapeutic activity.

Given all our observations, our hypothesis to explain the discrepancies between our uptake results and the survival results are:

1. PLA-HPG NPs are the only formulation that does not trigger an immune response (i.e. activation of microglia and/or recruitment of TAM – see **Fig 4g in the manuscript**), especially compared to the PLA-PEG NPs. This likely reduces their clearance from the brain space, which allows to compensate a decreased internalization.
2. From all the formulations, PLA-HPG NPs demonstrated higher specificity toward tumor cells (see **Fig 6d in the manuscript**): although their overall uptake is reduced, a specific internalization by tumor cells might allow them to reach the efficacy of PLA NPs or PLA-HPG-CHO NPs.

3. It is possible that, for a very potent drug such as EB, the level of internalization of the PLA-HPG NPs was sufficient to reach the IC₅₀ of the drug. In that case, more internalization obtained with PLA or PLA-HPG-CHO NPs will not add any additional therapeutic activity. The PLA-PEG NPs presented a two-times lower absolute internalization by tumor cells at 4 h (see **Table s1 in the manuscript**), and a decreased drug release, compared to PLA-HPG NPs, which might be sufficient to decrease the intracellular drug concentration below its IC₅₀.

Another crucial point raised by the reviewer, is that the NPs might not have to be internalized by the cells to be efficient, and simply releasing the drug in the extracellular space could be sufficient. This is certainly at least partly true for a drug such as EB, however other class of drugs—such as genetic material or radiosensitizers—might require internalization to be efficient. That is why we still believe that understanding the cellular tropism of the different NPs formulations is of primary importance for a rational development of new delivery systems to be administered by CED.

These new drug release results have been added to the main text as Fig 8a.

Reviewer #2 (Remarks to the Author):

This paper explores some very important topic that is the distribution of nanoparticles in the brain depending of surface chemistry. The biomedical need behind is the need to cross the blood-brain-barrier, that is to date possible with only a very poor efficacy using most nano-platform developed. Investigation of convection enhanced delivery is a **very important** strategy, as this technology is now translated at the bedside, demonstrating efficacy in glioblastoma. The determination of the cell type specificity is also **very important** as the brain function is mainly governed by network and very specific cell types. Transfecting in the same location a different cell type can induce opposite effects. The nanoparticles used in this paper are PLA based nanoparticles and the mode of delivery is convection enhanced delivery. The main objective of this paper was to explore the transfection efficacy in connection with surface modifications such as « stealth » properties and bio-adhesive modification. PLA, PLA-PEG, PLA-HPG (avoiding phagocytic recognition) and PLA-HPG-CHO (bio-adhesive) were investigated. Moreover, an innovative in vitro cell culture test was implemented to predict transfection efficiency. **The quality of the experimental data is high with excellent statistical analysis.**

The main claim of this paper is that surface nanoparticles properties influence transfection efficacy as well as the cellular tropism of transfection and that it can be predict in vitro using the innovative test developed in this paper. Interestingly, stealth formulation decreases cellular uptake, while bio-adhesive surface modifications dramatically enhance cellular uptake. Correlated efficacy was demonstrated in a glioblastoma model, especially for PLA-HPG and PLA-HPG-CHO nanoparticles that are similarly efficient. Indeed, a low biocompatibility is observed for the adhesive nanoparticles, that induce a strong gliosis as well as a microglial activation and neutrophils infiltration.

In conclusion, this is an important work demonstrating for the first time relationship between surface properties of nanoparticles and transfection as well as cellular distribution inside the brain after convective infusion. The low impact of stealth modification for local delivery was not anticipated. The impact of adhesive modifications is new. A therapeutic efficacy is also demonstrated. **An important drawback in this study is the toxicity observed with the more efficient and innovative nanoparticle harboring adhesive properties inducing an inflammatory gliotic reaction.**

We thank the reviewer for his/her careful evaluation of our work and thoughtful comments. We agree that the toxicity of the most internalized formulation (PLA-HPG-CHO NPs) represent a significant drawback for its further use in the brain. **As described in our manuscript, this result highlights the fact that an increased internalization can found its limitation in the apparition of cellular toxicity.**

Reviewer #3 (Remarks to the Author):

This work investigates the cellular fate of PLA-based nanoparticles with various surface chemistries

injected into the brain. Internalization is enhanced with particles coated with HPG with aldehyde functionality, as also shown in their in vitro studies. These particles were used to delivery epothilone B in an orthotopic model of glioblastoma.

The work is quite thorough and includes particle characterization, distribution analysis after injection into the brain of both normal and tumor bearing mice by imaging and flow cytometry, and in vivo efficacy using these nanoparticles to deliver drug to a glioblastoma model. **Overall, I am quite impressed with the amount of work covered by the article.**

However, my major reservation is that there is not much new information provided to the scientific community from the studies. Much of their findings have been previously reported in various publications (although this study is certainly the most comprehensive compilation). The impact of the work is therefore not very high. Another concern is that some of the graphs particularly the pie charts misrepresent the data.

We thank the reviewer for his/her kind evaluation of our work and thoughtful comments. However, we disagree that our results—showing substantial differences in neural cell tropism among particles with different surface chemistry—do not contain new information. Although our intent was to build on past advances—using materials that are familiar and potentially translatable to clinical practice—to our knowledge, the role of nanoparticle surface chemistry in cellular tropism after CED has never been described in detail.

Some more specific comments are below:

1. The DiA could significantly alter the physicochemical characteristics of the particles. This dye also inserts into cell membranes and is used as cell tracking studies, which could further confound the studies.

Here, the reviewer raises the concerns that (1) the encapsulation of the dye might alter the physicochemical characteristics of the particles, and (2) the dye could be released from the particles, and thus the fluorescence measurement would not represent the presence of NP. These concerns have been carefully monitored for in our laboratory, since we have been using the Di family of dyes for many of our studies due to their compatibility with our particle systems and diverse fluorescence options.

- (1) Within our own study, we showed that the size, surface charge and particle morphology through TEM did not change significantly between the dye vs drug encapsulated NPs (in the manuscript: **Fig 1** and **Table s2**).
- (2) We have previously evaluated the retention of the Di dyes by performing in vitro release studies, showing that the dye is not escaping PLA based particles, even in the presence of very high concentrations of proteins (Deng Y. et al, PNAS 2016, 113(41): 11453-11458).

We have added this information to our manuscript.

2. Flow cytometry of brain homogenates is challenging and methods used need to be more thoroughly outline and validated, including actual dot plots from flow cytometry and details of methods such as removal of fat.

We agree with the reviewer that the method we describe has not been traditionally used to analyze NPs uptake. However, our methodology has been adapted from multiple studies in the neuroscience field, which fully and thoroughly validated it:

Spoelgen, R., et al. (2011). "A novel flow cytometry-based technique to measure adult neurogenesis in the brain." Journal of Neurochemistry **110**: 165-175.

Guez-Barber, D., et al. (2012). "FACS purification of immunolabeled cell types from adult rat brain." J Neurosci Methods **203**(1): 10-18.

Hayashi, Y., et al. (2011). "A novel, rapid, quantitative cell-counting method reveals oligodendroglial reduction in the frontopolar cortex in major depressive disorder." *Mol Psychiatry* 16(12): 1155-1158.

These references have been added to the method section.

Since we are isolating a single population and looking specifically at one color shift due to particles uptake, we thought that it was appropriate to show it as a histogram. However, to address the reviewer's concern regarding the methodology, we have added an example of the dot plot to show the staining and particle shift on a single plane (**Fig 4**) and to better describe our result analysis.

Fig 4: Example of dot plots on single planes obtained through data analysis. (a) The cell population is first gated from all the event collected, and plotted on the Cy-5 (y-axis, staining of cell markers) and DiA channels (x-axis, fluorescent NPs). **(b)** Representative results obtained for each staining in untreated brains (top row) and brains treated for 4 h with PLA-HPG-CHO NPs, showing how the different percentages of cells are extracted.

This new information have been added as Supp. Figure 3.

3. Figure 4 states that HPG and HPG-CHO particles do not ramify microglia but it looks like all microglia in these figures are ramified. Normal untreated brain needs to be shown and GFAP-stained images moved to main text.

As requested by the reviewer, we have added new pictures of an untreated brain as a supplementary figure (Supp Fig 6) and moved the GFAP-stained images to the main text (added to Fig 4).

We think that there was a misunderstanding regarding the figure caption of figure 4, and we apologize for it. Microglia are naturally found in a ramified state (see new Supp. Fig 6 showing the microglia staining of an untreated brain) and our experiments showed that while HPG and PEG particles kept the microglia ramified (new Fig 4j, k in the manuscript), the HPG-CHO and PLA particles activated the cells, changing their conformation into an amoeboid shape (new Fig 4i, l).

4. Figure 5: Glial population increases from day 7 to 8 and its not clear if this is due to nanoparticle treatment or tumor growth. Controls should be included.

Once again, we think that there was a misunderstanding regarding figure 5 and we apologize for it. The data presented in figure 5 relate to the composition of the brain in terms of cellular type, in the

absence of NPs, in order to compare the composition with and without an orthotopic tumor grown for 7 or 8 days. Hence, those brain compositions constitute the controls for the results obtained after NPs infusion, which are presented in figure 6.

5. In Figure 6, it appears the authors stained for Iba-1, GFAP, GFP, NeuN and then gated each cell marker based off DiA. Then took the number of positive DiA cells and the relative MFI for those positive cells and multiplied by the number of Iba-1 (or GFAP/NeuN or GFP) positive cells. Instead for readers it would be helpful to first gate off total cell number and tell the reader the total number of positive DiA cells within that total number. Then break down what percentage of that total DiA positive cells are Iba-1, GFAP, GFP and NeuN positive. Level of brightness can be shown in supplemental information.

The reviewer’s suggestion is very thoughtful and we spent a lot of time considering the most understandable way of presenting those results. We came up with this analysis because we are including the whole hemisphere in our tissue processing, while the particles are not present within the whole hemisphere but only a part of it. Hence, it was important for accuracy to isolate only the region where there were particles. Moreover, the purpose of our study was to show how the particles distribute amongst the different cell compartments, and not to explore how many cells were taking up particles, which was why we ended up analyzing and displaying the data this way. However, we reanalyzed some of our results to demonstrate that the different analysis methods lead to the same conclusions (**Fig. 5**).

Fig 5: Analysis comparison: Our original analysis (“Manuscript”) included calculating the total amount of fluorescence in each cell type and then multiplying it to the number of cells in a given population to get a total fluorescence per population. The sum of these values were set as a 100%, and then particle distribution within each population was displayed as %. In re-analysis #1, we display one of our original analysis of accounting for MFI but not the cell population. Re-analysis #2 represents data analyzed per the reviewer’s suggestion in taking the NP positive population and looking at how many of those cells are astrocytes, microglia, neurons and tumor cells. In both re-analysis methods, the numbers did not significantly vary from the original manuscript analysis (student’s t-test > 0.05).

Reviewers' Comments:

Reviewer #1 (Remarks to the Author)

The revised manuscript addresses several concerns/issues raised in the initial review. With the added analysis of perinuclear staining within the cells and the drug release data presented in the supplemental data, the manuscript is certainly more complete. I agree with the reviewer that particles that have been cleared to either the CSF or blood will be difficult to quantitate, and so the issue for me is not so much clearance of the NPs from the brain, but how much NP is present within the extracellular space or cellular matrix (ie. not in the cell). The approach used is great for capturing cell fluorescence, but what would the fluorescence in the non-cellular compartments be? As the authors note this is not easy to assess as the process for collecting the cells likely removes all non-cellular NPs. I don't believe this is going to be a huge fraction of the NPs injected, but I don't believe as the authors imply that the delivery to the cells is 100%. This concern is relatively minor, and should not significantly change the points of the present manuscript regarding surface properties of the NPs and uptake into the various cells in the brain.

The authors are commended for addressing the concerns of the initial review and the additional studies make the manuscript stronger.

Reviewer #2 (Remarks to the Author)

The responses as well as the modifications done by the authors introduce many improvements that enhance the signification of the paper.

The efficacy of the strategy for local vectorization is now deeply demonstrated in the glioblastoma model and the efficacy of the new chemistry particle well documented.

At the end, this is a referent paper that enlighten the power of local delivery technology in the field of vectorization. To my knowledge no equivalent paper was previously published.

The drawback of this strategy is the fact that increased internalization induces cellular toxicity, that limits its functional and therapeutic effect. I would add a paragraph discussing this item specifically. Too much nanomedicine papers described at the preclinical level miracle efficacy without any investigation of biocompatibility. But, it is probably very important to know that « too much vectorization » can be toxic toxic.

Reviewer #3 (Remarks to the Author)

The authors have thoughtfully addressed my concerns and I think the manuscript is now suitable for publication.

REVIEWERS' COMMENTS:

Reviewer #1 (Remarks to the Author):

The revised manuscript addresses several concerns/issues raised in the initial review. With the added analysis of perinuclear staining within the cells and the drug release data presented in the supplemental data, the manuscript is certainly more complete. I agree with the authors that particles that have been cleared to either the CSF or blood will be difficult to quantitate, and so the issue for me is not so much clearance of the NPs from the brain, but how much NP is present within the extracellular space or cellular matrix (ie. not in the cell). The approach used is great for capturing cell fluorescence, but what would the fluorescence in the non-cellular compartments be? As the authors note this is not easy to assess as the process for collecting the cells likely removes all non-cellular NPs. I don't believe this is going to be a huge fraction of the NPs injected, but I don't believe as the authors imply that the delivery to the cells is 100%. This concern is relatively minor, and should not significantly change the points of the present manuscript regarding surface properties of the NPs and uptake into the various cells in the brain.

The authors are commended for addressing the concerns of the initial review and the additional studies make the manuscript stronger.

We thank the reviewer. We certainly agree that a fraction of the administered particles might reside in the extracellular space and may never be internalized by the cells. When we mentioned 100% of delivery, we referred to 100% of delivery to the brain tissue (as opposed to systemic administration that allows a delivery to the brain tissue of only a few % in best cases), not to the delivery to the cells.

Reviewer #2 (Remarks to the Author):

The responses as well as the modifications done by the authors introduce many improvements that enhance the signification of the paper. The efficacy of the strategy for local vectorization is now deeply demonstrated in the glioblastoma model and the efficacy of the new chemistry particle well documented. At the end, this is a referent paper that enlighten the power of local delivery technology in the field of vectorization. To my knowledge no equivalent paper was previously published.

The drawback of this strategy is the fact that increased internalization induces cellular toxicity that limits its functional and therapeutic effect. I would add a paragraph discussing this item specifically. Too much nanomedicine papers described at the preclinical level miracle efficacy without any investigation of biocompatibility. But, it is probably very important to know that « too much vectorization » can be toxic toxic.

We thank the reviewer. We agree that one of the major conclusions of this study is the importance of balancing more internalization for more efficacy vs potential toxicity. As requested by the reviewer, we added a few sentences to the manuscript to emphasize this point.

Reviewer #3 (Remarks to the Author):

The authors have thoughtfully addressed my concerns and I think the manuscript is now suitable for publication.

We thank the reviewer.